# PU-Bench: A Unified Benchmark for Rigorous and Reproducible PU Learning

**Qiuyi Chen**[1]*, **Haiyang Zhang** [1]*†, **Leqi Zhang**[1], **Changchun Li**[2,3], **Jia Wang**[1], **Wei Wang**[1]

[1]School of Advanced Technology, Xi'an Jiaotong-Liverpool University, China
[2]College of Computer Science and Technology, Jilin University, China
[3]Key Laboratory of Symbolic Computation and Knowledge Engineering, Jilin University, China
`{qiuyi.chen2002,leqi.zhang23}@student.xjtlu.edu.cn`
`{haiyang.zhang,jia.wang02,wei.wang03}@xjtlu.edu.cn`
`changchunli93@gmail.com`

## Abstract

Positive-Unlabeled (PU) learning, a challenging paradigm for training binary classifiers from only positive and unlabeled samples, is fundamental to many applications. While numerous PU learning methods have been proposed, the research is systematically hindered by the lack of a standardized and comprehensive benchmark for rigorous evaluation. Inconsistent data generation, disparate experimental settings, and divergent metrics have led to irreproducible findings and unsubstantiated performance claims. To address this foundational challenge, we introduce **PU-Bench**, the first unified open-source benchmark for PU learning. PU-Bench provides: 1) a unified data generation pipeline to ensure consistent input across configurable sampling schemes, label ratios and labeling mechanisms; 2) an integrated framework of 18 state-of-the-art PU methods; and 3) standardized protocols for reproducible assessment. Through a large-scale empirical study on 8 diverse datasets (**2880** evaluations in total), PU-Bench reveals a complex yet intuitive performance landscape, uncovering critical trade-offs between effectiveness and efficiency, and systematically mapping method robustness against variations in label frequency and selection bias. It is anticipated to serve as a foundational resource to catalyze reproducible, rigorous, and impactful research in the PU learning community. The source code is publicly available at `https://github.com/XiXiphus/PU-Bench`.

## 1 Introduction

Positive-Unlabeled (PU) learning tackles a common classification problem where only some positive examples are labeled and the unlabeled rest are a mixture of unidentified positives and true negatives (Elkan & Noto, 2008; Bekker & Davis, 2020). This setting arises frequently in real-world applications where negative examples are difficult or costly to annotate. For example, in recommender systems, it is known which items users like, but not which they dislike (Zhou et al., 2021; Zhang et al., 2021). This characteristic makes PU learning an essential technique across diverse fields, and the past few years have seen a rapid development in algorithm design for PU learning, such as disease-related gene identification (Yang et al., 2012; Molaei & Jalili, 2025), drug-drug interaction prediction (Zheng et al., 2019), document retrieval (Wang et al., 2024; Zhang et al., 2024) and medical image classification (Nagaya & Ukita, 2021).

Despite these advances, a significant challenge remains due to the lack of a standardized, unified, and comprehensive benchmark for a fair comparison among different algorithms. This issue causes two major obstacles to further development. First, there is a remarkable inconsistency in experimental setups and evaluation protocols across studies. Researchers often employ different datasets, varying data sampling strategies (e.g., case-control vs. single-training-set) (Bekker & Davis, 2020; Mielniczuk & Wawrzeńczyk, 2024), and divergent labeling assumptions (e.g., selected completely

---

*Equal contribution.
†Corresponding author.

at random vs. selected at random) (Gong et al., 2021; 2025), which inevitably leads to inconsistent and incomparable results. As indicated in Appendix Table C.1, these disparate settings are rather common, preventing a fair and holistic understanding of algorithmic performance. Second, the validity of performance claims is often undermined by the high sensitivity of PU methods to empirical factors. Our empirical results confirm that variations in the label ratio or labeling assumption can be significant enough to alter relative performance rankings of state-of-the-art algorithms. Given that these factors are not uniformly controlled in prior work, many published comparisons may not reflect the true capabilities of individual methods, leading to potentially unreliable conclusions.

To address these challenges and establish a solid foundation for future research, we propose **PU-Bench**, to the best of our knowledge, the first comprehensive, open-source, and unified benchmark for PU learning. This work makes three primary contributions: 1) **Unified Open-Source Benchmarking Framework**: We design and release an open-source, modular framework for the rigorous and reproducible evaluation of PU learning algorithms. It features a configurable PU generator, a unified training pipeline, and a comprehensive evaluation suite to ensure fair and consistent comparisons. 2) **Comprehensive Empirical Study**: We conduct the most comprehensive empirical study in PU learning to date, benchmarking 18 representative methods across 8 diverse datasets. Our experimental design covers 20 configurations per method-dataset pair, including 14 label ratios under SCAR and 3 SAR mechanisms evaluated at 2 representative label frequencies. This protocol yields a total of **2,880** controlled evaluations, enabling rigorous assessment of both peak performance and robustness across the full spectrum of PU learning scenarios. 3) **In-Depth Analysis and Actionable Guidelines**: We provide extensive evaluation and analysis from various perspectives, including effectiveness, efficiency and complexity, as well as robustness to varying label ratios and selection bias. Our findings reveal the strengths and limitations of the current PU methods, and propose a set of practical, data-driven guidelines for algorithm selection and design.

## 2 PRELIMINARIES

**Problem Setup.** In PU learning, an observed training sample consists of a set of labeled positives $\mathcal{LP}$ and an unlabeled set $\mathcal{U}$ that mixes positives and negatives. The standard one-sided labeling assumption posits that only positives can be labeled, i.e., $p(Y = 1|S = 1) = 1$ (equivalently, $p(S = 1|Y = 0) = 0$) (Elkan & Noto, 2008; Bekker & Davis, 2020). Let $\pi = p(Y = 1)$ denote the class prior. The class-conditional densities are given by $f_+(x) = p(x \mid Y = 1)$ and $f_-(x) = p(x \mid Y = 0)$. The marginal is the mixture $f(x) = \pi f_+(x) + (1 - \pi)f_-(x)$. The objective is to learn a decision function $h_\theta : \mathcal{X} \to [0, 1]$ that estimates the posterior $p(Y = 1|X = x)$. Key notations used throughout this paper are summarized in Appendix Table A.1.

**Data Sampling Scheme.** Training data can be typically generated under two sampling schemes: the **single-training-set (ss)** scenario and the **case-control (cc)** scenario (Bekker & Davis, 2020). In the ss scenario, the training set is created by drawing samples i.i.d. from the population distribution, and only the positive examples within this set have a chance of being labeled. The cc scenario assumes that the labeled-positive set is drawn i.i.d. from $p(x \mid Y = 1)$ and the unlabeled set is drawn i.i.d. from the population distribution $p(x)$. The primary distinction between the two scenarios is the composition of the unlabeled sample. Under cc, the unlabeled data follow the population mixture of the class-conditional distributions $p(x \mid Y = 1)$ and $p(x \mid Y = 0)$ with mixing proportion $\pi$, whereas the ss scenario uses a mixture with a different effective proportion because labeled positives are excluded from $\mathcal{U}$ (Mielniczuk & Wawrzeńczyk, 2024). Consequently, PU methods should always state and consider the sampling scheme when conducting experiments and interpreting results. A detailed illustration of the differences between ss and cc can be found in Appendix C.2.

**Labeling Mechanisms.** Under both sampling schemes, there is one set of samples drawn i.i.d. from the population distribution $p(x)$ and one drawn i.i.d. from the positive class $p(x \mid Y = 1)$, governed by a labeling mechanism with propensity $e(x) = p(S = 1 \mid Y = 1, X = x)$ (Bekker & Davis, 2020). In PU learning, the most widely used assumption is **Selected Completely At Random (SCAR)**, which posits a constant propensity, i.e., $e(x) = c$ (Bekker & Davis, 2020). A more general assumption is **Selected At Random (SAR)**, where the propensity depends on features, i.e., $e(x) \neq c$ (Bekker & Davis, 2018).

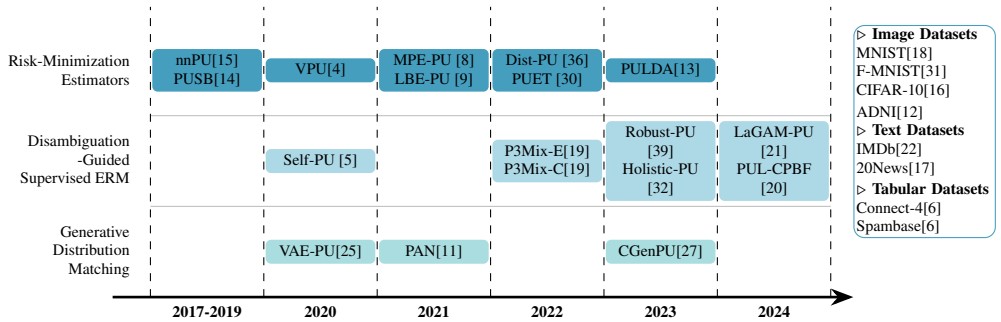

Figure 1: Overview of the collected methods and datasets.

## 3 PU-BENCH

In this section, we introduce PU-Bench, a unified benchmark designed for PU learning. We first detail the scope of our study, covering the datasets and collected algorithms (Section 3.1), followed by a detailed explanation of the system architecture that ensures reproducible data generation, model training, and evaluation (Section 3.2).

### 3.1 DATASETS AND METHODS

**Datasets.** PU-Bench evaluates 8 widely used datasets spanning three different modalities: *text* - **IMDb** (movie reviews for sentiment classification) (Maas et al., 2011) and **20News** (text classification) (Lang, 1995); *image* - **MNIST** (handwritten digit classification) (LeCun et al., 1998), **Fashion-MNIST** (F-MNIST) (Xiao et al., 2017), **CIFAR-10** (natural image classification) (Krizhevsky & Hinton, 2009), and **ADNI** (structural MRI for Alzheimer's disease diagnosis) (Jack Jr et al., 2008); *tabular* - **Spambase** (email spam detection) (Dua & Graff, 2019), and **Connect-4** (board game outcome prediction) (Dua & Graff, 2019). This diverse selection of tasks and domains ensures a comprehensive assessment of model robustness and generalizability. Detailed descriptions of each dataset are provided in Appendix B.1.

Although these datasets have been widely adopted in PU learning, existing works often construct PU data in disparate ways, such as adopting different label ratios, making varied assumptions about the unlabeled data distribution (e.g., SCAR or SAR), and employing distinct sampling designs (e.g., ss vs. cc). These inconsistencies result in heterogeneous experimental settings and limit the reproducibility of reported results. To address this issue, PU-Bench provides a systematically organized collection of datasets and a standardized PU data generation pipeline, ensuring consistent input across methods and allowing researchers to focus on methodological development rather than data preparation.

**Methods.** PU-Bench has implemented a total number of 18 established PU learning algorithms. The selection is based on methodological relevance, reproducibility, and general applicability. Specifically, we prioritize influential and recent PU methods from top-tier venues, focusing exclusively on domain-agnostic methods with publicly available implementations or author-provided implementations. Methods with unavailable code and non-reproducible pipelines are excluded. A detailed summary of the selected methods is provided in Appendix Table B.3 and Fig. 1.

We categorize the collected methods into three groups based on their algorithmic strategies, as illustrated in Fig. 1: 1) *Risk-Minimization Estimators*: methods that directly minimize empirical risk or its variants under PU constraints, including **nnPU** (Kiryo et al., 2017), **PUSB** (Kato et al., 2019), **VPU** (Chen et al., 2020a), **LBE-PU** (Gong et al., 2021), **MPE-PU** (Garg et al., 2021), **PUET** (Wilton et al., 2022), **Dist-PU** (Zhao et al., 2022), **PULDA** (Jiang et al., 2023); 2) *Disambiguation-Guided Supervised ERM*: methods that first resolve label ambiguity in the unlabeled pool by constructing pseudo-labels or selecting proxy negatives/positives (often under class-prior constraints or with group/meta signals), and then a standard supervised ERM model is trained on $\mathcal{LP} \cup \mathcal{U}$. This training phase often leverages mixup, consistency regularization, or iterative teacher-student self-training. This group includes **Self-PU** (Chen et al., 2020b), **P3Mix** (Li et al.,

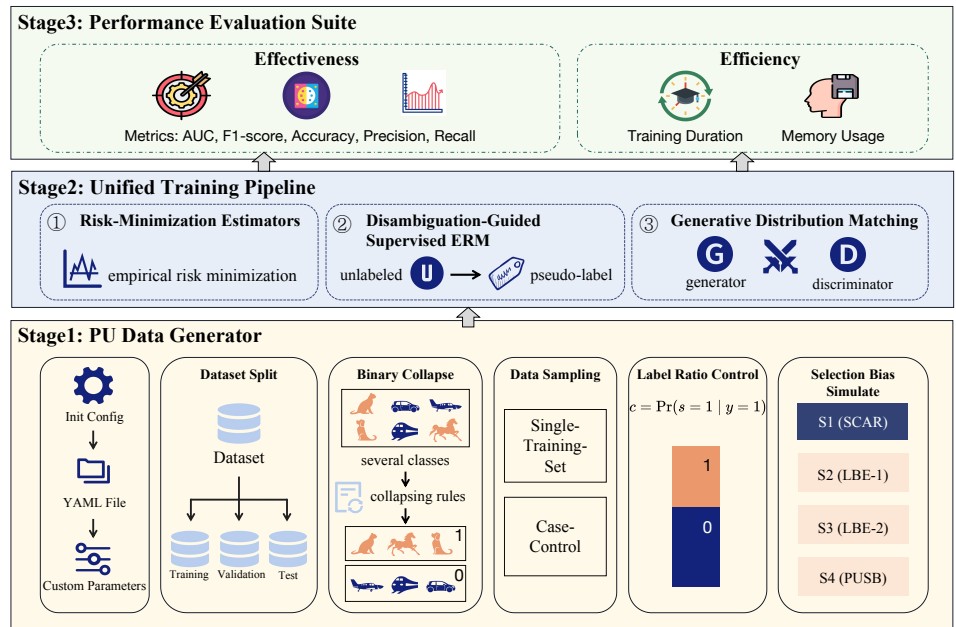

Figure 2: The modular framework for PU-Bench.

2022), **Robust-PU** (Zhu et al., 2023), **Holistic-PU** (Xinrui et al., 2023), **LaGAM-PU** (Long et al., 2024), **PUL-CPBF** (Li et al., 2024); 3) *Generative Distribution Matching*: methods that align positive and unlabeled distributions with model-induced predictions via generative or adversarial modeling, including **PAN** (Hu et al., 2021), **VAE-PU** (Na et al., 2020) and **CGenPU** (Papič et al., 2023). This taxonomy provides a structured framework for analyzing methodological differences. Detailed descriptions of each method are provided in Appendix B.2.

## 3.2 SYSTEM DESIGN

Despite the extensive research on PU learning, the field currently lacks a standardized framework for empirical evaluation, posing a challenge to experimental reproducibility and fair comparison. This methodological inconsistency manifests across the entire experimental pipeline. Studies vary widely in their data generation processes: employing different sampling schemes, e.g. ss or cc (Mielniczuk & Wawrzeńczyk, 2024), varying the size of the positive set, and operating under distinct labeling assumptions, e.g. SCAR or SAR (Gong et al., 2021), as detailed in Appendix C.1. Furthermore, disparities in training protocols, hyperparameter optimization, and the choice of evaluation metrics (shown in Appendix C.4) make it difficult to aggregate findings or reliably observe the true state-of-the-art. To address these critical challenges, PU-Bench aims to facilitate rigorous, reproducible, and transparent evaluation of PU learning algorithms. It is built upon three core, interoperable components that standardize the experimental workflow: 1) a PU Data Generator for generating experimental conditions, 2) a Unified Training Pipeline for consistent model execution, and 3) a Performance Evaluation Suite for comprehensive analysis and comparison, as shown in Fig. 2.

**PU Data Generator.** The PU Data Generator is responsible for systematically transforming standard classification datasets into diverse and reproducible PU learning scenarios. The generation process follows a structured, multi-stage pipeline. It begins with a binarization module that converts multi-class datasets into a binary Positive-Negative (PN) format, which is then split into training, validation, and test sets, with the total number of samples in the training set $N$ and class prior $\pi$ fixed. Then, the generator simulates the data collection environment by choosing one data sampling scheme to define where the labeled samples $\mathcal{LP}$ are from; details of the implementation of the two data sampling schemes are shown in Appendix C.2. Finally, the labeling simulation is performed by specifying the value of label ratio $c$, which controls how many $\mathcal{LP}$ are sampled, and the labeling mechanism, which defines the selection strategy for $\mathcal{LP}$. Our framework supports multiple mech-

anisms to emulate real-world complexities, including (**S1**) the common SCAR assumption, where every positive has a uniform labeling probability; (**S2**) the instance-dependent sampling strategy where the propensities are based on the auxiliary posterior $\hat{p}(x)$ to favor high-posterior positives (Gong et al., 2021); (**S3**) another instance-dependent sampling strategy where the propensities emphasize ambiguous or boundary positives (Gong et al., 2021); (**S4**) the posterior sharpening strategy where top-scoring positives are deterministically selected under a sharpened posterior (Kato et al., 2019). Details of the propensity functions and scoring rules are given in Appendix C.3.

**Unified Training Pipeline.** This module currently supports 18 state-of-the-art PU methods, as detailed in Sec. 3.1. To ensure rigorous reproducibility and eliminate confounding variables, PU-Bench employs a fully modular, configuration-driven training framework where all algorithms are instantiated from external YAML descriptors. These descriptors specify model backbones, PU loss functions, and shared hyper-parameters, including optimizer type, learning-rate schedules, and weight initialization. The framework accommodates multiple data modalities via specialized encoders for text, image, and tabular data. Training begins by loading the configuration and PU-formatted datasets, after which the corresponding PU learning criterion and learner are instantiated. A unified trainer automates forward/backward passes, loss computation, metric logging, and checkpointing. By allowing parameter adjustments through simple YAML edits, the framework ensures streamlined, reproducible training, evaluation, and visualization across all PU methods.

**Performance Evaluation Suite.** The Performance Evaluation Suite is designed to provide a holistic and statistically robust assessment of each algorithm, measuring performance across two primary dimensions: predictive effectiveness and computational efficiency. This dual focus addresses a critical gap in the literature, where inconsistent reporting undermines the fairness of cross-method comparisons and obscures the practical trade-offs of different approaches. To ensure a standardized evaluation, we implement a unified protocol where all metrics are computed on a held-out, ground-truth test set. For effectiveness, we record 5 widely used evaluation metrics including accuracy (Acc), precision, recall, macro-F1, area under the ROC curve (AUC). To measure efficiency, the framework logs wall-clock time and peak GPU memory per epoch; a checkpoint is written whenever validation-set macro-F1 hits a new best, and the full config, seeds, metric traces, and hardware stats are archived for full reproducibility.

## 4 EXPERIMENTAL RESULTS

This section presents the primary empirical results of our study, where we benchmark 18 representative PU learning methods across 8 datasets. To establish a controlled and comparable baseline that aligns with established practices, all experiments are conducted under a conventional PU setting. This protocol, the most widely adopted configuration in prior studies, simulates the case-control sampling scenario, assumes the SCAR labeling mechanism, and utilizes a fixed label frequency of $c = 0.1$. Detailed statistics of the simulated PU datasets, specific implementation settings such as the backbone models employed, and expanded set of performance metrics are provided in Appendix D for full reproducibility and deeper analysis. To assess whether the observed accuracy gaps are robust to random seed variability rather than artifacts of sampling noise, we additionally perform two-sided paired $t$-tests with Holm–Bonferroni correction between each PU method and nnPU (Kiryo et al., 2017) across all datasets; the detailed procedure and corrected $p$-values are summarized in Appendix D.2.2.

### 4.1 EFFECTIVENESS COMPARISON

To establish a practical performance ceiling and contextualize the results, we include a fully supervised PN classifier trained with complete label information as an oracle reference. According to the results presented in Table 1, Fig. 3 and Appendix D.2.1, we have the following observations:

**Risk-Minimization Estimators perform strongly but with higher variance.** As shown in Table 1, while this family often achieves the best results on specific datasets (e.g., Dist-PU on ADNI and CIFAR-10, LBE-PU on MNIST and F-MNIST), it suffers from significantly higher variance and threshold sensitivity than disambiguation-guided counterparts, exemplified by LBE-PU's collapse on ADNI under extreme positive-label imbalance (Appendix D.2.3). Within this group, nnPU

Table 1: Accuracy score of all PU methods across all datasets under the conventional setting. The best results are shown in **bold** and the second best are underlined. Entries marked with * are significantly better according to the two-sided paired $t$-tests in Appendix D.2.2.

| Type | Method | IMDb | 20News | MNIST | F-MNIST | CIFAR-10 | ADNI | Connect-4 | Spambase |
|---|---|---|---|---|---|---|---|---|---|
| Risk-Minimization Estimators | nnPU [15] | 77.37±4.82 | **88.81**±3.81 | 94.85±0.09 | 96.67±1.43 | 85.30±2.63 | 65.75±4.02 | 74.71±2.33 | 81.66±2.73 |
| | PUSB [14] | 77.86±3.82 | 87.49±4.05 | 95.19±1.35 | 95.76±1.62 | 87.68*±2.85 | 65.85±4.35 | 85.07±2.55 | 81.56±2.95 |
| | VPU [4] | 78.08*±4.13 | 87.76±4.33 | 96.46*±1.52 | 97.50±1.83 | 83.76±3.12 | 66.31±4.72 | 85.62*±2.82 | 79.25±3.22 |
| | MPE-PU [8] | 77.65±1.93 | 86.68±4.55 | 95.90±0.67 | 97.79±2.03 | 83.41±3.35 | 65.47±5.05 | 74.60±3.05 | 81.11±3.45 |
| | LBE-PU [9] | 76.25±4.73 | 85.74±4.81 | **97.23***±0.08 | **98.42***±2.22 | 83.98±3.62 | 65.75±5.42 | 83.94±3.31 | 68.53±3.71 |
| | PUET [30] | 68.41±5.32 | 86.49±5.31 | 95.08±2.12 | 97.77±2.63 | 76.23±4.11 | 71.56*±6.13 | 84.23±3.83 | 84.43±4.23 |
| | Dist-PU [36] | 77.88*±5.02 | 88.65±5.05 | 95.70±0.39 | 95.31±2.42 | **88.09***±3.85 | **75.02***±5.75 | 73.85±3.55 | 85.71*±3.95 |
| | PULDA [13] | 75.30±1.47 | 86.67±0.03 | 96.69±0.35 | 96.92±0.42 | 87.02*±0.68 | 67.91±5.45 | 78.18*±4.48 | 84.80±2.30 |
| Disambiguation-Guided Supervised ERM | Self-PU [5] | 74.05±3.23 | 86.93±3.46 | 90.86±1.02 | 93.22±1.31 | 76.73±2.43 | 66.41±4.21 | 75.67±2.51 | 72.10±2.92 |
| | P3Mix-E [19] | **78.27***±3.53 | 84.42±3.23 | 94.26±1.15 | 96.92±1.52 | 87.36±2.65 | 69.21±4.55 | 84.79±2.75 | 84.14±3.15 |
| | P3Mix-C [19] | 77.48*±3.83 | 88.02±3.92 | 95.23±1.32 | 96.53±1.72 | 87.65±2.91 | 67.69*±4.92 | 85.10±2.72 | 81.46±3.42 |
| | Robust-PU [39] | 77.37±4.12 | 88.81±4.15 | 95.05±1.45 | 97.71±1.93 | 84.30±3.19 | 67.91±5.25 | 74.71±3.27 | 81.66±3.62 |
| | Holistic-PU [32] | 64.40±4.41 | 75.20±4.42 | 92.75±1.61 | 62.16±2.43 | 57.08±3.41 | 65.07±5.62 | 79.80±3.51 | 56.59±3.93 |
| | LaGAM-PU [21] | 76.81±4.71 | 84.90±4.65 | 95.03±0.89 | 97.69±0.07 | 86.22±3.60 | 63.64±5.05 | 85.08*±3.72 | **86.77***±4.15 |
| | PUL-CPBF [20] | 75.26±5.01 | 84.92±4.91 | 88.31±1.92 | 98.07*±0.52 | 80.27±3.93 | 66.11±6.31 | **88.12***±4.02 | 85.94*±4.41 |
| Generative Distribution Matching | VAE-PU [25] | 66.48±4.02 | 77.51±3.91 | 76.56±1.51 | 61.29±7.71 | 49.24±5.81 | 50.38±1.52 | 79.39±2.72 | 59.15±3.11 |
| | PAN [11] | 70.26±8.41 | 78.20±4.25 | 87.45±1.71 | 93.81±1.95 | 68.70±6.12 | 61.99±4.91 | 78.37±3.06 | 38.71±8.40 |
| | CGenPU [27] | 70.85±3.89 | 86.08±4.61 | 86.56±1.92 | 93.35±4.66 | 57.79±9.42 | 60.45±5.31 | 79.42±3.32 | 81.66±3.72 |
| | PN | 79.89±0.83 | 92.32±0.02 | 96.54±0.92 | 98.94±0.82 | 94.88±0.57 | 82.01±0.31 | 92.38±0.78 | 91.03±0.66 |

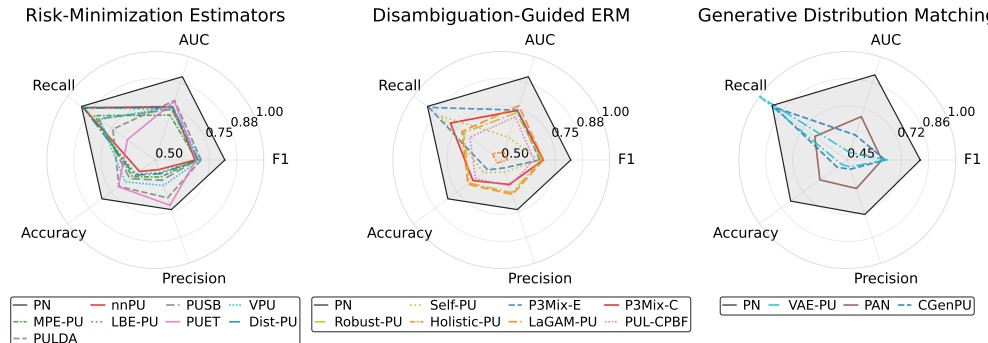

Figure 3: Performance of PU methods on the ADNI dataset across five evaluation metrics.

and Dist-PU emerge as the most consistent performers. In contrast, VPU frequently shifts between recall-dominant and precision-dominant behaviors depending on the dataset (recall-leaning on ADNI, precision-leaning on CIFAR-10). The radar visualizations in Fig. 3 and Appendix Fig. 10 underscore this instability. Despite extreme outliers like VPU's recall spikes or Self-PU's erratic shapes, nnPU remains a remarkably resilient baseline, maintaining a balanced configuration across diverse modalities.

**Disambiguation-guided methods provide balanced and stable performance.** While rarely achieving the absolute highest accuracy scores (Table 1), this family delivers the most reliable overall stability across modalities. Within this family, the two P3Mix variants lead and complement each other: P3Mix-E pushes recall under heavy contamination (notably on ADNI, IMDb, and vision tasks), while P3Mix-C trades some recall for higher precision. Robust-PU is a stable low-variance baseline, especially competitive on text (20News), and LaGAM-PU performs strongly on tabular data (Spambase); by contrast, Holistic-PU clearly trails across datasets. Consistent with Fig. 3, Fig. 9, and Fig. 11, this family produces balanced profiles over five metrics, reflecting well-rounded decision behavior. The main outliers are P3Mix-E's recall spikes with lower precision on ADNI/CIFAR-10 and Holistic-PU's jagged high-recall/low-precision shapes on vision and tabular data, but these remain infrequent and do not undermine the family's overall stability.

**Generative Distribution Matching methods lag behind overall.** From Table 1, this family consistently underperforms across modalities and rarely enters the top two. While PAN and CGenPU achieve moderate success on text data (e.g., 20News), they fail to generalize to vision (e.g., CIFAR-

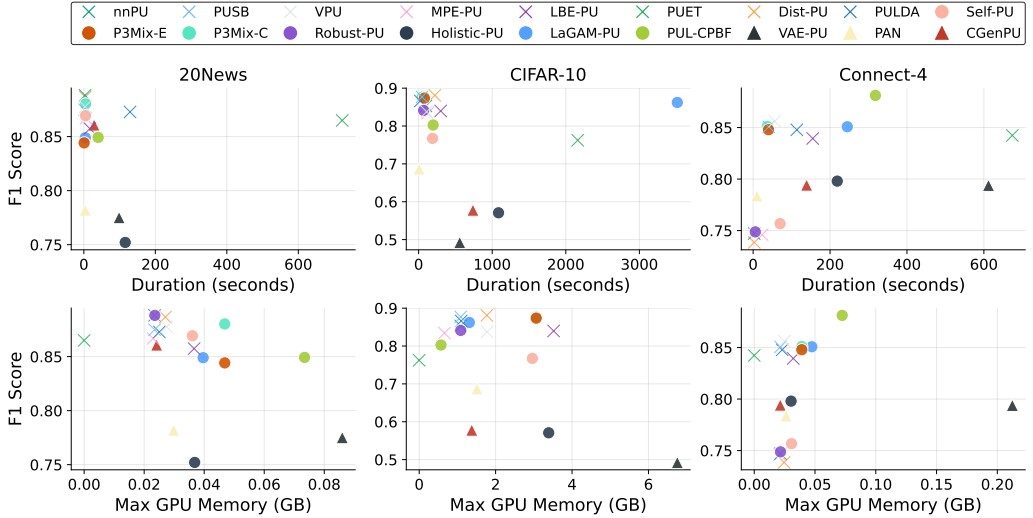

Figure 4: Time and space analysis of PU methods on three representative datasets (20News, CIFAR-10 and Connect-4).

10) and medical imaging (ADNI). VAE-PU is the weakest and least stable estimator overall. This brittleness is exemplified by PAN's critical failure on Spambase, where discriminator-induced reliability collapse produces extreme recall-precision imbalance (Appendix D.2.3). As shown in the radar charts (Appendix Fig. 11), profiles are compact and jagged, indicating a severe imbalance between F1, AUC, precision, and recall. Although rare outliers exist, such as the balanced profile of CGenPU on 20News, they are insufficient to offset the group's broader lack of robustness.

**Performance is Highly Contingent on Data Modality.** A critical finding of our benchmark is that no single method emerges as a universal winner. The best method depends heavily on data modality. For instance, LBE-PU achieves state-of-the-art accuracy on simple images (even surpassing supervised PN on MNIST) but degrades on complex ones like ADNI. In contrast, VPU and P3Mix variants are more stable, delivering consistently strong though not always top-tier performance across a wider range of data types. Surprisingly, nnPU, despite its simplicity, remains highly competitive and often outperforms newer methods. This suggests that progress in the field is not always linear, and some earlier principles remain highly robust. These findings stress the need for comprehensive benchmarking and careful alignment of method choice with problem characteristics.

### 4.2 COMPUTATIONAL ANALYSIS

Fig. 4 illustrates the critical trade-off between predictive effectiveness (F1 score) and computational costs (training duration and GPU memory usage) across the selected datasets of diverse modalities. The results reveal substantial variation in the efficiency profiles of the benchmarked methods:

**Training time.** Foundational methods such as nnPU and PUSB are exceptionally efficient, completing epochs in seconds on 20News and MNIST, and staying close to the lower bound on CIFAR-10. This efficiency is directly attributable to their relatively simple objective formulations, which avoid complex generative or adversarial components. Dist-PU, VPU, and P3Mix introduce a moderate computational overhead. While slower than the simplest baselines, they balance advanced modeling components with efficient implementation. Their cost-effectiveness stems from building upon lightweight extensions, such as distributional losses, mixup strategies, or simple pseudo-labeling rather than fundamentally altering the training pipeline. In contrast, PUL-CPBF, Holistic-PU, and VAE-PU are the most computationally demanding. Their prolonged epoch durations are a direct consequence of their complex, multi-stage architectures. These pipelines involve iterative processes like clustering, meta-learning, or adversarial training, which require multiple forward/backward passes or the optimization of auxiliary models, incurring substantial computational costs.

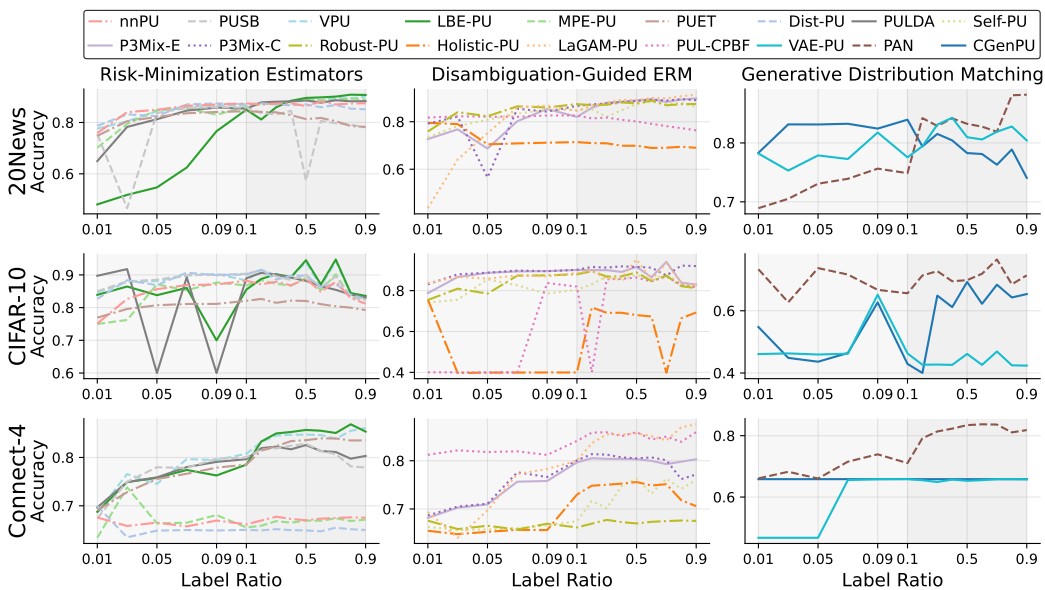

Figure 5: Effectiveness of PU methods with respect to different label ratio.

**Memory consumption.** A clear divide in memory usage exists, directly linked to architectural complexity. A large group of methods, including nnPU, PUSB, VPU, Robust-PU, and Dist-PU, are highly memory-efficient ($< 1$ GB) as they rely on a standard single-classifier pipeline. In contrast, methods with more complex designs have a significantly larger memory footprint. VAE-PU is the most demanding (up to 7-8 GB) due to its generative components (encoder-decoders, discriminators), while others like P3Mix-E and PUL-CPBF increase memory by incorporating mechanisms like EMA teachers and multi-view data processing.

**Trade-off analysis.** A group of methods, notably VPU, Self-PU and Dist-PU, achieves a strong balance, e.g., high F1 with short training time and low memory overhead. P3Mix-C/E yield comparably high F1 scores at modest extra cost. In contrast, Robust-PU and MPE-PU yield only competitive performance while requiring more substantial resources. Established baselines like nnPU and PUSB are exceptionally efficient, requiring the least training times. However, this comes at the expense of a modest reduction in F1 score compared to the top-tier methods. In contrast, Holistic-PU, VAE-PU and PUL-CPBF demonstrate a less favorable trade-off. These methods are the most computationally intensive yet often yield lower or less stable F1 scores, severely limiting their practical applicability.

## 5 FURTHER ANALYSIS

In this section, we conduct a deeper investigation into the robustness of the benchmarked methods. We systematically vary two experimental conditions that can reflect the quality of labeled data: the size of labeled samples controlled by label ratio and the underlying labeling mechanism controlled by propensity score.

### 5.1 IMPACT OF LABEL FREQUENCY ON MODEL PERFORMANCE

We vary the value of label ratio $c$ from 0.01 to 0.09 with a step of 0.02 and from 0.1 to 0.9 with a step of 0.1. Results on selected datasets are shown in Fig. 5 and Appendix D.3, revealing distinct patterns of stability and label efficiency across the three families.

**Overall performance trends.** Most methods within the *Risk–Minimization Estimation* and *Disambiguation-Guided ERM* categories demonstrate remarkable label efficiency. Their perfor-

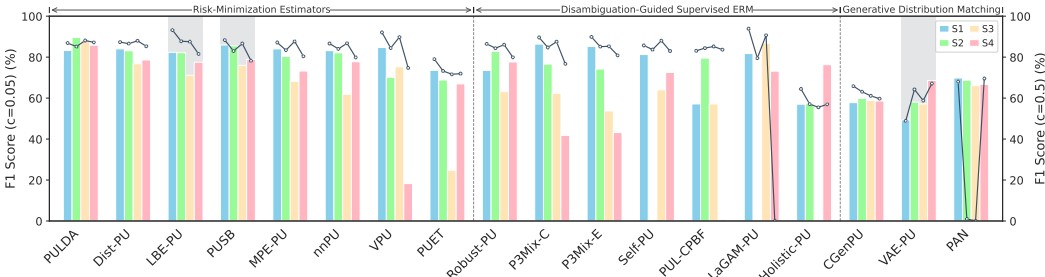

Figure 6: F1 scores of PU methods under different **labeling assumptions** on CIFAR-10. The bars show the performance in a low-label regime ($c = 0.05$), while the lines indicate performance in a high-label regime ($c = 0.5$). Methods with grey background are those especially designed for selection bias.

mance curves typically show a rapid initial ascent at low label ratios (e.g. $c < 0.1$) before gracefully saturating, indicating a strong capability to learn effectively from limited labeled examples. For instance, VPU and P3Mix-C approach their peak performance on CIFAR-10 with $c$ ranging only from 0.03 to 0.05. In contrast, *Generative Distribution Matching* methods exhibit high sensitivity to both label ratio and data modality. Their performance curves are often erratic. Some methods show competitive performance with extremely small labeled sets (e.g. VAE-PU and CGenPU on image and text data), demonstrating their potential to deal with scarce data. However, this behavior does not generalize to all modalities. More importantly, these methods often exhibit poor scalability, showing little improvement as the label ratio $c$ grows. Such limitations likely stem from the inherent instability of optimizing complex generative objectives under the sparse and ambiguous signals characteristic of PU data.

**Analysis of notable performers and outliers.** The analysis reveals several distinct performance profiles: 1) Efficiency: VPU and P3Mix variants are exceptionally effective in low-label regimes, demonstrating high label efficiency by achieving competitive accuracy with minimal supervision; 2) Scalability: LBE-PU exhibits notable late-stage gains, improving substantially as more labeled data becomes available, suggesting it scales well with supervision; 3) Instability: methods like PUL-CPBF and Holistic-PU show high performance variance, with performance fluctuating significantly at low label ratios; 4) Saturation: Finally, methods like VAE-PU remain at a performance bottleneck, showing little improvement even as labeled data becomes plentiful, indicating that their primary limitations are architectural rather than related to data scarcity.

## 5.2 ROBUSTNESS TO SELECTION BIAS

To assess robustness to labeling bias, we evaluate all methods under three realistic SAR variants, in contrast to the standard SCAR setting. Experiments were conducted at both a low ($c = 0.05$) and a high ($c = 0.5$) label ratio, with results shown in Fig. 6 and Appendix D.4.

**Low-$c$ regime ($c = 0.05$).** When labeled data is scarce, the choice of labeling mechanism has a profound impact. Under the standard SCAR assumption, top-tier methods, e.g. VPU, Dist-PU, and P3Mix-C, maintain their dominance, while the selection-bias-aware methods like PUSB and LBE-PU remain competitive. However, when the mechanism shifts to SAR, a universal performance degradation is observed. The *Risk-Minimization Estimation* group demonstrates strong robustness across different labeling assumptions, with no method suffering from catastrophic performance failure. Crucially, methods explicitly designed for bias mitigation (i.e., PUSB, LBE-PU) exhibit superior resilience, incurring smaller performance penalties than their SCAR-assuming counterparts. A notable anomaly is VAE-PU, which paradoxically achieves higher accuracy under SAR than SCAR, though its overall performance remains uncompetitive. Ultimately, these results confirm that the benefit of bias-aware modeling is prominent in the low-label regime.

**High-$c$ regime ($c = 0.5$).** When labels are plentiful ($c = 0.5$), most methods converge to high F1 scores, and the performance gap between SCAR and SAR settings narrows significantly, though it

does not vanish. Consistent with the low-c regime, the *Risk-Minimization Estimation* group continues to demonstrate high stability. Crucially, we find that robust SCAR learners like VPU can outperform dedicated SAR-aware methods in this high-data regime. This finding highlights a critical interaction between label ratio and selection bias, implying that the optimal strategy for handling SAR conditions may depend on the amount of labeled data available.

**Practical Takeaways.** These findings yield a clear practical recommendation. When label acquisition is suspected to be non-random, particularly when labeled data is sparse, employing a bias-aware method like PUSB or LBE-PU is highly preferable. Conversely, in scenarios where labels are plentiful and the mechanism approaches near-SCAR conditions, lightweight and robust methods with semi-supervised regularization (VPU, P3Mix) offer state-of-the-art performance.

## 6    CONCLUSION AND FUTURE DIRECTIONS

**Conclusion.**    This work addresses the absence of a standardized benchmark in PU learning by introducing the open-source framework, **PU-Bench**. Through this framework, we conducted the largest systematic evaluations to date in this field, analyzing 18 representative algorithms across an extensive range of datasets and conditions. The results demonstrate that the performance of PU learning methods is highly context-dependent, varying significantly with data modality, label frequency, and labeling mechanism. As a foundational empirical study, this work provides the necessary grounding to guide future theoretical and algorithmic advancements. We anticipate **PU-Bench** to serve as a cornerstone for the community, accelerating research by providing a rigorous, standardized toolkit for comparing the state-of-the-art and inspiring next-generation solutions.

**Future directions.**    Through our empirical analysis, we identify the following future directions: 1) **Rigorous evaluation**: Our benchmark reveals that more recent and complex methods can be outperformed, at least on some datasets, by simpler baselines (i.e., nnPU, VPU). This underscores the necessity for future research to conduct more rigorous, standardized evaluations against these strong, efficient baselines to validate claims of novelty and improvement. 2) **Real-world stress test**: A key vulnerability exposed by our study is the poor performance of most methods under severe data constraints (e.g., high label sparsity and selection bias), as characterized by many real-world applications. This reveals a significant gap between current algorithmic capabilities and practical needs. Therefore, a critical direction for future research is the design of methods that are inherently robust to the challenges of extremely limited and biased supervision.

## ACKNOWLEDGMENT

This research was supported by the Collaborative Research Project (RDS10120240248) at Xi'an Jiaotong-Liverpool University. Additionally, this research has received the financial support provided by the Fundamental Research Funds for the Central Universities, Jilin University.

## REPRODUCIBILITY

We provide full configuration files and scripts to reproduce all experiments. All results are deterministically reproducible from fixed random seeds: we set and log seeds for dataset splits, data loaders, model initialization, and CUDA/cuDNN backends. Each experiment is specified by a single YAML configuration and a seed; re-running with the same seed exactly reproduces the reported metrics. Detailed packaging and usage instructions are provided in Appendix F.

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

APPENDIX

TABLE OF CONTENTS

# A  NOTATIONS

Table A.1: Summary of notation. Uppercase letters (e.g., $X$) denote random variables; lower-case (e.g., $x$) denote realizations. Densities (e.g., $f(x)$) are evaluated pointwise; expectations (e.g., $\mathbb{E}_{X \sim f}[g(X)]$) are taken with respect to the distribution of $X$.

| Symbol | Description |
|---|---|
| $X$ | Feature vector, $X \in \mathcal{X}$ |
| $Y$ | Latent class label $\{0, 1\}$ (positive$= 1$) |
| $S$ | Labeling indicator $\{0, 1\}$ (labeled$= 1$) |
| $\mathcal{D}$ | Observed training sample $\{(x_i, s_i)\}_{i=1}^{N}$ |
| $\mathcal{P}, \mathcal{N}$ | Sets of positives and negatives in the underlying PN dataset |
| $\mathcal{LP}$ | $\{\, x_i : (x_i, s_i) \in \mathcal{D},\ s_i = 1 \,\}$ (labeled positives) |
| $\mathcal{U}$ | $\{\, x_i : (x_i, s_i) \in \mathcal{D},\ s_i = 0 \,\}$ (unlabeled; mixture of positives and negatives) |
| $\mathcal{U}_P$ | Unlabeled positives, $\mathcal{U}_P := \mathcal{U} \cap \mathcal{P}$ |
| $N$ | Training set size used to construct PU splits |
| $N_\ell$ | Target size of the labeled-positive set $\mathcal{LP}$ |
| $N_u$ | Size of the unlabeled set in the PU training split |
| $N_v$ | Validation set size |
| $N_p^{tr}, N_n^{tr}$ | Positive/negative counts in the PN training set |
| $N_p^{te}, N_n^{te}$ | Positive/negative counts in the test set |
| $\pi$ | Class prior, $p(Y = 1)$ |
| $e(x)$ | Labeling propensity, $p(S = 1 | Y = 1, X = x)$ |
| $c$ | Label frequency, $p(S = 1 | Y = 1) = \mathbb{E}[e(X) | Y = 1]$ |
| $f(x)$ | Marginal density of $X$ |
| $f_+(x), f_-(x)$ | Positive density, $p(x | Y = 1)$ and negative density, $p(x | Y = 0)$ |
| $f_\ell(x), f_u(x)$ | Labeled-positive density, $p(x | S = 1)$ and unlabeled density, $p(x | S = 0)$ |
| $h_\theta(x)$ | Estimate of $p(Y = 1 | X = x)$ |
| $\hat{p}(x)$ | Estimated posterior $p(Y{=}1 \mid X{=}x)$ |
| $\sigma(z)$ | Logistic sigmoid, $\sigma(z) = \frac{1}{1+e^{-z}}$ |
| $\mathbf{w}$ | Parameter vector in the logistic-regression example for $\hat{p}(x) = \sigma(\mathbf{w}^\top x)$ |

# B  DATASET AND METHODS

## B.1  DETAILS OF DATASETS

**IMDb** (Maas et al., 2011) is a sentiment analysis dataset containing 50,000 movie reviews from the Internet Movie Database, evenly split into positive and negative labels. Reviews are preprocessed to remove HTML tags and are tokenized for text classification tasks.

**20News** (Lang, 1995) is a text classification dataset consisting of approximately 18,000 newsgroup documents evenly distributed across 20 topics. Each document is represented as a bag-of-words vector, with labels corresponding to its topic category.

**MNIST** (LeCun et al., 1998) contains 70,000 grayscale images of handwritten digits (0-9), each of size 28×28 pixels. The dataset is split into 60,000 training and 10,000 testing images for digit classification.

**Fashion-MNIST** (Xiao et al., 2017) consists of 70,000 28×28 grayscale images of fashion items across 10 classes, we also use F-MNIST to refer to Fashion-MNIST in this paper.

**CIFAR-10** (Krizhevsky & Hinton, 2009) comprises 60,000 32×32 color images evenly divided into 10 object classes. Each class contains 6,000 images, split into 50,000 training and 10,000 test samples.

**ADNI** (Jack Jr et al., 2008) (Alzheimer's Disease Neuroimaging Initiative) is a medical imaging dataset containing structural MRI scans for Alzheimer's disease research. Each scan is labeled according to the patient's clinical diagnosis.

**Spambase** (Dua & Graff, 2019) contains 4,601 email samples represented by 57 numerical features, such as word frequency and character frequency. Labels indicate whether each email is spam or not.

**Connect-4** (Dua & Graff, 2019) is a dataset derived from the board game "Connect Four." It contains 67,557 game states represented by 42 categorical features corresponding to the positions on a $6 \times 7$ board. Each position can take one of three values: x (first player's piece), o (second player's piece), or b (blank). Labels indicate the outcome of the game from the perspective of the first player: win, loss, or draw.

Table B.1: Class labels and their indices for multi-class datasets.

| Dataset | Classes (index) |
|---|---|
| 20News | Alt (0), Comp (1), Misc (2), Rec (3), Sci (4), Soc (5), Talk (6) |
| IMDb | Negative (0), Positive (1) |
| MNIST | 0, 1, 2, 3, 4, 5, 6, 7, 8, 9 |
| F-MNIST | T-Shirt/Top (0), Trouser (1), Pullover (2), Dress (3), Coat (4), Sandal (5), Shirt (6), Sneaker (7), Bag (8), Ankle Boot (9) |
| CIFAR-10 | Airplane (0), Automobile (1), Bird (2), Cat (3), Deer (4), Dog (5), Frog (6), Horse (7), Ship (8), Truck (9) |
| Alzheimer MRI | NonDemented (0), VeryMildDemented (1), ModerateDemented (2), MildDemented (3) |
| Connect-4 | Loss (0), Win (1), Draw (2) |
| Spambase | Not Spam (0), Spam (1) |

Table B.2: PU learning datasets and statistics with index-based class mapping. $N_p^{tr}, N_n^{tr}$ denote the number of positive and negative samples in the training set, respectively; $N_p^{te}, N_n^{te}$ for the test set; $N_v$ for validation size.

| Dataset | Pos. VS Neg.(Indices) | Input Size | Train $(N_p^{tr}, N_n^{tr})$ | Test $(N_p^{te}, N_n^{te})$ | Validation | Total Size |
|---|---|---|---|---|---|---|
| 20News | 0,1,2,3 **VS.** 4,5,6 | 384 | $(6,326, 4,874)$ | $(4,254, 3,278)$ | $0.01 (N_v = 114)$ | 18,846 |
| IMDb | 1 **VS.** 0 | 384 | $(12,375, 12,375)$ | $(12,500, 12,500)$ | $0.01 (N_v = 250)$ | 50,000 |
| MNIST | 0,2,4,6,8 **VS.** 1,3,5,7,9 | $28 \times 28$ | $(29,197, 30,203)$ | $(4,926, 5,074)$ | $0.01 (N_v = 600)$ | 70,000 |
| F-MNIST | 0,2,3,4,6 **VS.** 1,5,7,8,9 | $28 \times 28$ | $(29,700, 29,700)$ | $(5,000, 5,000)$ | $0.01 (N_v = 600)$ | 70,000 |
| CIFAR-10 | 0,1,8,9 **VS.** 2,3,4,5,6,7 | $32 \times 32 \times 3$ | $(19,800, 29,700)$ | $(4,000, 6,000)$ | $0.01 (N_v = 500)$ | 60,000 |
| ADNI | 0 **VS.** 1,2,3 | $128 \times 128$ | $(2,552, 2,516)$ | $(622, 658)$ | $0.01 (N_v = 52)$ | 6,400 |
| Connect-4 | 1 **VS.** 0,2 | 126 | $(35,222, 18,282)$ | $(8,895, 4,617)$ | $0.01 (N_v = 541)$ | 67,557 |
| Spambase | 1 **VS.** 0 | 57 | $(1,435, 2,208)$ | $(363, 558)$ | $0.01 (N_v = 37)$ | 4,601 |

## B.2 DETAILS OF IMPLEMENTED METHODS

**nnPU** (Kiryo et al., 2017) introduces a non-negative risk estimator for PU learning. Unlike unbiased PU learning where the empirical risk can become negative during optimization, nnPU constrains the risk to remain non-negative by truncating any negative component to zero. This modification prevents the model from overfitting to noise when training flexible models such as deep neural networks with limited positive data.

**PUSB** (Kato et al., 2019) operates under the invariance of order assumption, which posits that the class posterior and the labeling probability induce the same ordering on the input space. It estimates the density ratio between labeled positive and unlabeled data either by minimizing a pseudo classification risk or through direct density ratio estimation. The classifier is then constructed by ranking instances according to this density ratio and setting the decision threshold at the precision-recall breakeven point.

**VPU** (Chen et al., 2020a) introduces a variational principle that avoids class-prior estimation by directly minimizing the KL divergence between the positive data distribution and the distribution induced by the classifier. The method optimizes a variational loss computed as the difference between the log expectation of the classifier output over unlabeled data and the expectation of the log

Table B.3: Summary of the collected methods.

| Type | Method | Repositories | Reference |
|------|--------|--------------|-----------|
| Risk-Minimization Estimators | nnPU [15] | `https://github.com/kiryor/nnPUlearning` | NeurIPS-2017 |
| | PUSB [14] | `https://github.com/MasaKat0/PUlearning` | ICLR-2019 |
| | VPU [4] | `https://github.com/HC-Feynman/vpu` | NeurIPS-2020 |
| | MPE-PU [8] | `https://github.com/acmi-lab/PU_learning` | NeurIPS-2021 |
| | LBE-PU [9] | Unpublished, required by us | IEEE TPAMI-2022 |
| | PUET [30] | `https://github.com/jonathanwilton/PUExtraTrees` | NeurIPS-2022 |
| | Dist-PU [36] | https://github.com/Ray-rui/Dist-PU | CVPR-2022 |
| | PULDA [13] | `https://github.com/jiangyangby/PULDA` | IEEE TPAMI-2023 |
| Disambiguation-Guided Supervised ERM | Self-PU [5] | `https://github.com/TAMU-VITA/Self-PU` | ICML-2020 |
| | P3Mix [19] | Available upon request | ICLR-2022 |
| | Robust-PU [39] | `https://github.com/woriazzc/Robust-PU` | KDD-2023 |
| | Holistic-PU [32] | `https://github.com/wxr99/HolisticPU` | NeurIPS-2023 |
| | LaGAM-PU [21] | `https://github.com/llong-cs/LaGAM` | CVPR-2024 |
| | PUL-CPBF [20] | Unpublished, required by us | ICML-2024 |
| Generative Distribution Matching | VAE-PU [25] | `https://github.com/byeonghu-na/vae-pu` | CIKM-2020 |
| | PAN [11] | `https://github.com/morning-dews/PAN` | AAAI-2021 |
| | CGenPU [27] | `https://github.com/apapich/CGenPU` | ESWA-2023 |

output over positive data. To handle limited positive data and prevent overfitting, VPU incorporates a MixUp-based consistency regularization that enforces prediction smoothness on interpolated samples between labeled positives and unlabeled data.

**MPE-PU** (Garg et al., 2021) addresses PU learning through two coordinated components: Best Bin Estimation for mixture proportion estimation and Conditional Value Ignoring Risk for classification. BBE estimates the proportion of positives in unlabeled data by leveraging a Positive-versus-Unlabeled classifier to identify a high-score bin containing a concentrated subset of positives, then computing the ratio of sample fractions in that bin. CVIR trains the classifier by iteratively removing the estimated proportion of unlabeled samples with the highest classification loss and training on the remaining samples as provisional negatives. The $TED^n$ framework alternates between updating the proportion estimate via BBE and refining the classifier via CVIR until convergence.

**LBE-PU** (Gong et al., 2021) addresses instance-dependent PU learning where the probability of labeling a positive example depends on its features, known as labeling bias. It formulates the problem via a graphical model characterizing the relationships among input features, latent true labels, and observed labeling status. The method jointly estimates the labeling bias function and the classifier parameters through maximum likelihood estimation, which is optimized via the Expectation-Maximization algorithm with gradient-based updates applicable to both linear (logistic) and deep (MLP) model instantiations.

**PUET** (Wilton et al., 2022) introduces a recursive greedy risk minimization framework for learning decision trees from PU data. It reinterprets tree learning as directly minimizing PU-based risk estimators (uPU or nnPU) rather than traditional impurity measures like Gini or entropy. The method constructs PU Extra Trees by recursively partitioning nodes to maximize the PU risk reduction, where each split is selected based on closed-form risk estimates computed from the positive and unlabeled subsets at that node. Additionally, the framework provides a risk reduction importance metric that directly quantifies each feature's contribution to minimizing the empirical risk on PU data.

**Dist-PU** (Zhao et al., 2022) addresses the negative-prediction preference inherent in cost-sensitive PU methods—where classifiers tend to over-predict the negative class as training progresses—by introducing a label distribution alignment perspective. It enforces consistency between the expected value of predicted labels and the ground-truth class prior over the unlabeled data, thereby globally constraining the proportion of negative predictions. To avoid trivial solutions where predictions uniformly concentrate around the class prior, the method incorporates entropy minimization to encourage confident predictions and Mixup regularization to mitigate confirmation bias and smooth decision boundaries.

**PULDA** (Jiang et al., 2023) extends the label distribution alignment framework by introducing a margin-based formulation that enhances instance-wise discriminability. Beyond aligning the expectation of predicted labels with the ground-truth distribution, it incorporates functional margins via

confidence penalization terms that push model outputs away from ambiguous regions near the decision boundary, preventing under-confident solutions. Furthermore, PULDA unifies this objective with a class prior estimation process (using BBE) to eliminate reliance on oracle priors, and employs an exponential moving average (EMA) based stochastic optimization algorithm that achieves a provable convergence rate under standard assumptions.

**Self-PU** (Chen et al., 2020b) integrates self-paced learning, self-calibrated loss reweighting, and teacher-student self-distillation for positive-unlabeled classification. The self-paced component progressively mines high-confidence unlabeled examples into a dynamically expanding trusted set, employing an in-and-out mechanism to iteratively update soft pseudo-labels for both positive and negative classes with balanced sampling rates. For remaining unconfident unlabeled data, a meta-learning based self-calibrated instance-aware loss adaptively reweights the combination of cross-entropy with soft labels and non-negative PU risk, constrained by a balancing factor to regulate supervision quality. The framework further enforces consistency regularization through collaborative distillation between two student networks trained with asynchronous learning paces (different sampling ratios), and between each student and its corresponding teacher network maintained via exponential moving average of weights.

**P3Mix** (Li et al., 2022) addresses decision boundary deviation in positive-unlabeled learning through heuristic mixup augmentation of a disambiguation-free objective. Treating unlabeled instances as pseudo-negative, it identifies marginal pseudo-negative instances (those exhibiting ambiguous predictive scores within a thresholded range) and selects their mixup partners exclusively from a candidate pool of positive instances with high prediction entropy (near-boundary positives), while other instances select partners randomly from the full training set. This generates augmented instances with partially positive soft labels that push the decision boundary toward the fully supervised position. The method proposes two robust variants: P3Mix-E incorporates early-learning regularization using mean-teacher estimated auxiliary targets to prevent memorization of imprecise supervision, while P3Mix-C performs pseudo-negative correction by reassigning high-confidence positive predictions to the positive class prior to mixup.

**Robust-PU** (Zhu et al., 2023) initializes the model via nnPU pre-training and executes an iterative three-stage training strategy. In the hardness measurement stage, it computes classification losses for all samples using temperature-scaled logistic or sigmoid loss, treating positive samples with positive ground truth and unlabeled samples as negatives. The sample weighting stage maps these hardness values to sample weights via the minimizer function of SPL-IR-Welsch regularization, where the mapping threshold is dynamically controlled by a training scheduler implementing linear, convex, concave, or exponential pacing functions to gradually relax selection criteria over iterations. The weighted supervised training stage treats unlabeled data as negative and performs binary cross-entropy optimization with the computed sample weights for multiple epochs. This iterative process progressively incorporates harder samples to prevent noise accumulation from early misclassification.

**Holistic-PU** (Xinrui et al., 2023) employs balanced resampling of positive and unlabeled data into equally-sized batches, initially treating unlabeled samples as negatives during training. It records the prediction score trajectory for each unlabeled sample across training iterations. A trend score is computed for each sample using a robust mean estimator applied to ordered differences between prediction scores at different time steps, capturing the tendency of negative samples to exhibit consistently decreasing scores while positive samples display chaotic or initially increasing patterns. The unlabeled data is then partitioned into pseudo-positive and pseudo-negative sets using Fisher's Natural Break algorithm based on the trend score distribution. Finally, a supervised classifier is trained on the resulting pseudo-labeled dataset.

**PUL-CPBF** (Li et al., 2024) establishes that the probability boundary of the asymmetric disambiguation-free risk is governed by the asymmetric penalty. The method first trains a set of weak classifiers with diverse probability boundaries by minimizing asymmetric disambiguation-free empirical risks under specific penalty values, forming a probability boundary fence. For each unlabeled instance, it locates the class posterior probability within the fence-defined range and generates a stochastic label via uniform sampling, then trains a strong classifier via self-training with consistency regularization.

**LaGAM-PU** (Long et al., 2024) employs hierarchical contrastive learning to extract latent group semantics through unsupervised clustering alignment, dichotomized cut-off based on binary predictions, and local neighbor smoothing. It then performs meta-learning-based label disambiguation via bi-level optimization: an inner loop virtually trains the model with current pseudo-labels, and an outer loop refines pseudo-labels using projected gradients with exponential moving average to minimize evaluation loss on a support set, alternating between label updates and classifier training.

**VAE-PU** (Na et al., 2020) proposes a generative approach to positive-unlabeled learning that eliminates the Selected Completely At Random assumption. The method employs a variational autoencoder architecture featuring two distinct latent representations: one capturing label information and another encoding observation indicators. Virtual positive-unlabeled instances are synthesized by combining the label-related latent factors derived from labeled positive examples with the observation-related factors extracted from unlabeled data. The optimization objective integrates the evidence lower bound with an adversarial alignment loss that matches generated samples to unlabeled data distributions and a label consistency loss that ensures generated instances exhibit positive-class characteristics. Training alternates between updating the generative network to produce informative pseudo-examples and optimizing the classifier using these synthesized instances.

**PAN** (Hu et al., 2021) introduces a predictive adversarial framework where a classifier assumes the role of a generator to identify likely positive instances from unlabeled data against a discriminator. The approach optimizes a divergence-based objective comprising three components: a classification term training the discriminator to recognize known positive examples, an alignment term encouraging prediction consistency between the classifier and discriminator on unlabeled data, and a symmetry correction term ensuring balanced gradient updates for positive and negative instances. This formulation enables training without requiring class-prior knowledge as input. The system alternates between discriminator updates using standard classification losses and classifier updates employing a policy-gradient-style mechanism based on discriminator feedback.

**CGenPU** (Papič et al., 2023) proposes a conditional GAN-based framework for positive-unlabeled learning that employs a generator, discriminator, and auxiliary classifier in a single-stage three-player minimax game. The training objective combines adversarial loss with an auxiliary loss that minimizes distribution divergence between labeled and generated positives while maximizing separation between generated positive and negative samples. An anchoring term stabilizes early training by enforcing high confidence on labeled positives. This approach simultaneously learns both class distributions without requiring prior knowledge of class priors or labeled negative examples, addressing the instability and architectural complexity of prior multi-generator methods.

# C  DATA GENERATION SETTINGS

## C.1  SETTINGS IN EXISTING PU WORK

Table C.1 provides a comprehensive summary of the data generation protocols employed in recent PU learning literature. For each method, we categorize the experimental setup based on three critical factors. First, we identify the data sampling scenario, distinguishing between the case-control (cc) and single-training-set (ss) paradigms. Second, we note whether the authors demonstrated robustness by systematically varying the label frequency-the proportion of true positives that are labeled (indicated by vary c). Finally, we specify the core labeling mechanism (e.g., SCAR, SAR) assumed in their experiments. For studies where these configurations were not explicitly stated, we inferred the settings by meticulously analyzing the described methodology and, where available, the associated source code to ensure an accurate and consistent comparison.

Table C.1: Data generation settings of the existing PU learning methods. cc denotes case-control and ss denotes single-training-set. "Varying $c$" indicates that the corresponding work demonstrates robustness by using different values of $c$.

| Method | Datasets | ss/cc | Varying $c$ | SCAR/SAR |
|---|---|---|---|---|
| nnPU[15] | CIFAR-10, MNIST, 20News, Epsilon | cc | No | SCAR |
| PUSB[14] | CIFAR-10, MNIST, Mushrooms, Spambase, Shuttle, Page Blocks, USPS, Connect-4, SwissProt | cc | Yes | SAR |
| VPU[4] | CIFAR-10, F-MNIST, STL-10, Page Blocks, Grid Stability, Avila | cc | Yes | SCAR |
| MPE-PU[8] | CIFAR-10, MNIST, IMDb | Not Given | No | Not Given |
| LBE-PU[9] | USPS, UCI: Australian, Madelon, Phishing, Vote, Banknote, Breast, HockeyFight, SwissProt | ss | Yes | SAR |
| PUET[30] | CIFAR-10, MNIST, IMDb | cc | No | SCAR |
| Dist-PU[36] | CIFAR-10, F-MNIST, ADNI | cc | Yes | SCAR |
| PULDA[13] | CIFAR-10, F-MNIST, ADNI, Tiny-ImageNet | cc | Yes | SCAR |
| Self-PU[5] | CIFAR-10, MNIST, ADNI | cc | No | SCAR |
| P3Mix[19] | CIFAR-10, F-MNIST, STL-10, Credit Card Fraud | cc | No | SCAR |
| Robust-PU[39] | CIFAR-10, MNIST, F-MNIST, STL-10, ADNI, UCI: Mushrooms, Shuttle, Spambase | cc | No | SCAR |
| Holistic-PU[32] | CIFAR-10, F-MNIST, STL-10, ADNI, Credit Card Fraud | cc | Yes | SCAR |
| LaGAM-PU[21] | CIFAR-10, STL-10, CIFAR-100, ADNI | ss | No | SCAR |
| PUL-CPBF[20] | CIFAR-10, F-MNIST, STL-10, ADNI | cc | No | SCAR |
| VAE-PU[25] | CIFAR-10, MNIST, 20News | cc | Yes | SAR |
| PAN[11] | CIFAR-10, MNIST, IMDb, 20News | ss | Yes | SCAR |
| CGenPU[27] | CIFAR-10, MNIST | cc | Yes | SCAR |

## C.2   SINGLE-TRAINING-SET AND CASE-CONTROL

**Single-training-set.**   This scenario preserves the original training set's size and composition. The number of positives to be labeled is $N_\ell = \lfloor c|\mathcal{P}| \rfloor$. These are drawn *without replacement* from $\mathcal{P}$ to form $\mathcal{LP}$. The remaining samples, $(\mathcal{P} \setminus \mathcal{LP}) \cup \mathcal{N}$, constitute the unlabeled set $\mathcal{U}$.

**Case-control.**   This scenario first selects $\mathcal{LP} \subseteq \mathcal{P}$ according to the chosen labeling strategy (Appendix C.3). The unlabeled set is then defined as the entire training pool, $\mathcal{U} = \mathcal{P} \cup \mathcal{N}$, i.e., $\mathcal{LP}$ is returned to $\mathcal{U}$. Consequently, $\mathcal{LP} \subseteq \mathcal{U}$ and $|\mathcal{U}| = N$, and the class mixture in $\mathcal{U}$ is identical to that of the full population. When enumerating indices, include all negatives and as many positives as needed to reach $|\mathcal{P}|$, prioritizing positives not in $\mathcal{LP}$ and reusing $\mathcal{LP}$ if necessary.

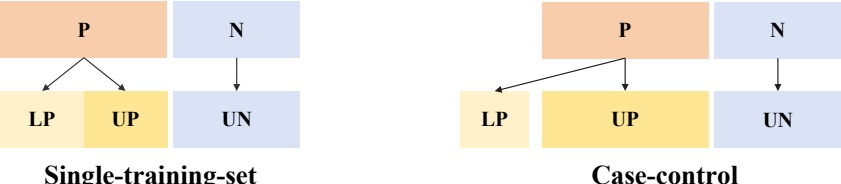

**Single-training-set**          **Case-control**

Figure 7: Comparison between the single-training-set and case-control. In both cases, negatives ($Y = 0$) are never labeled ($S = 1$), so all negatives in the training pool are assigned to the unlabeled set $\mathcal{U}$. In the single-training-set case, the labeled positives $\mathcal{LP}$ are sampled without replacement from $\mathcal{P}$ and removed from $\mathcal{U}$, yielding $(\mathcal{P} \setminus \mathcal{LP}) \cup \mathcal{N}$, whereas in the case-control scenario, $\mathcal{LP}$ is sampled from $\mathcal{P}$ but then returned so that $\mathcal{U} = \mathcal{P} \cup \mathcal{N}$ follows the class mixture of the overall training data distribution.

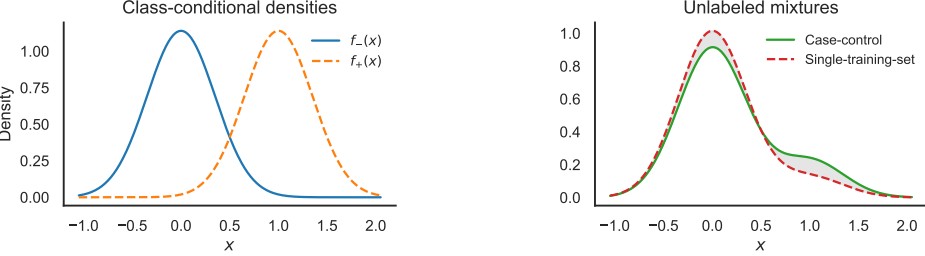

Figure 8: One-dimensional Gaussian toy example illustrating how single-training-set and case-control affect only the unlabeled density. The class-conditional densities $f_+(x)$ and $f_-(x)$ are fixed Gaussians with distinct means (left), while the unlabeled density $f_u(x)$ changes its effective class mixture under single-training-set versus case-control (right). Under case-control, $f_u$ matches the overall training population mixture, whereas under single-training-set, removing a fraction of positives into $\mathcal{LP}$ reduces the positive mass in $\mathcal{U}$ and shifts $f_u(x)$ closer to $f_-(x)$.

### C.3 LABELING STRATEGIES

The details of the four labeling strategies evaluated under our framework are:

**S1 (SCAR).** Select $N_\ell$ instances uniformly without replacement from $\mathcal{P}$ to form $\mathcal{LP}$. This corresponds to a constant labeling propensity $e(x) = c$ for all $x \in \mathcal{P}$.

**S2 (LBE Strategy 1).** Using the auxiliary posterior $\hat{p}(x)$ (cf. Table A.1 for $\sigma$ and $\mathbf{w}$), define instance-dependent propensities

$$e_1(x) = \left[\hat{p}(x)\right]^k, \quad k > 1.$$

Normalize $\pi_1(x) = \frac{e_1(x)}{\sum_{x' \in \mathcal{P}} e_1(x')}$ and select $\mathcal{LP}$ by weighted sampling without replacement of size $N_\ell$ using probabilities $\pi_1(x)$, which favors high-posterior positives; in our experiments we set $k = 10$ (Gong et al., 2021).

**S3 (LBE Strategy 2).** Using the same $\hat{p}(x)$, define

$$e_2(x) = \left[1 - \hat{p}(x)\right]^k, \quad k > 1.$$

Normalize $\pi_2(x) = \frac{e_2(x)}{\sum_{x' \in \mathcal{P}} e_2(x')}$ and select $\mathcal{LP}$ by weighted sampling without replacement of size $N_\ell$ using probabilities $\pi_2(x)$. This scheme favors boundary or otherwise ambiguous positives; in our experiments we also set $k = 10$ (Gong et al., 2021).

**S4 (PUSB Posterior Sharpening).** Using the same $\hat{p}(x)$, construct sharpened scores

$$s_\alpha(x) = \left[\hat{p}(x)\right]^\alpha, \quad \alpha \gg 1.$$

Rank all $x \in \mathcal{P}$ in descending order of $s_\alpha(x)$ and deterministically take the top-$N_\ell$ as $\mathcal{LP}$. This rule concentrates labeling on the highest-posterior positives; in our experiments we use $\alpha = 20$ (Kato et al., 2019).

### C.4 EVALUATION METRICS FOR PU LEARNING METHODS

For a fair and comprehensive evaluation, we compile the metrics reported in the original papers of each algorithm in Tab. C.2. We adopt AUC, F1, Accuracy, Precision, and Recall as our primary measures. Together, these metrics offer complementary perspectives on performance across diverse experimental conditions. Below, we briefly introduce each metric used to assess PU-learning methods.

**AUC.** The Area Under the ROC Curve (AUC) (Bradley, 1997) assesses a classifier's ability to rank positive instances above negative ones, independently of any fixed threshold. It can be viewed as the probability that a randomly selected positive instance receives a higher score than a randomly selected negative instance. AUC is widely regarded as a robust, threshold-free metric for binary classification.

**F1-score.** The F1-score (Powers, 2011) is the harmonic mean of Precision and Recall, combining them into a single measure. It is defined as

$$\text{F1} = \frac{2 \cdot \text{Precision} \cdot \text{Recall}}{\text{Precision} + \text{Recall}}$$

where $\text{Precision} = \frac{TP}{TP+FP}$ and $\text{Recall} = \frac{TP}{TP+FN}$. The F1-score is particularly useful on imbalanced datasets, as it penalizes extreme disparities between Precision and Recall, yielding a more balanced assessment of classification performance.

**Accuracy.** Accuracy is the proportion of correctly classified instances among all samples:

$$\text{Accuracy} = \frac{TP + TN}{TP + TN + FP + FN}$$

where $TP$, $TN$, $FP$, and $FN$ denote true positives, true negatives, false positives, and false negatives, respectively. Although accuracy is simple and intuitive, it can be misleading under severe class imbalance, where it is dominated by the majority class.

**Precision.** Precision (Powers, 2011) measures the proportion of true positives among all predicted positives:

$$\text{Precision} = \frac{TP}{TP + FP}$$

where $TP$ and $FP$ denote true positives and false positives, respectively. High precision indicates few false positive errors.

**Recall.** Recall (Powers, 2011) quantifies the proportion of true positives among all actual positives:

$$\text{Recall} = \frac{TP}{TP + FN}$$

where $FN$ represents false negatives. High recall indicates that the model captures most of the positive samples.

Table C.2: Evaluation metrics reported in their original papers ($\checkmark$ = reported, — = not reported).

| Method | AUC | F1 | Accuracy | Precision | Recall |
|---|---|---|---|---|---|
| nnPU[15] | — | — | — | — | — |
| PUSB[14] | — | — | — | $\checkmark$ | $\checkmark$ |
| VPU[4] | $\checkmark$ | — | $\checkmark$ | — | — |
| LBE-PU[9] | — | — | $\checkmark$ | — | — |
| MPE-PU[8] | — | — | $\checkmark$ | — | — |
| PUET[30] | — | $\checkmark$ | $\checkmark$ | — | — |
| Dist-PU[36] | $\checkmark$ | $\checkmark$ | $\checkmark$ | $\checkmark$ | $\checkmark$ |
| PULDA[13] | $\checkmark$ | $\checkmark$ | $\checkmark$ | $\checkmark$ | $\checkmark$ |
| Self-PU[5] | — | — | $\checkmark$ | — | — |
| P3Mix[19] | — | — | $\checkmark$ | $\checkmark$ | $\checkmark$ |
| Robust-PU[39] | — | — | $\checkmark$ | — | — |
| Holistic-PU[32] | $\checkmark$ | $\checkmark$ | $\checkmark$ | $\checkmark$ | $\checkmark$ |
| LaGAM-PU[21] | $\checkmark$ | $\checkmark$ | $\checkmark$ | — | — |
| PUL-CPBF[20] | $\checkmark$ | $\checkmark$ | $\checkmark$ | $\checkmark$ | $\checkmark$ |
| VAE-PU[25] | — | — | $\checkmark$ | — | — |
| PAN[11] | — | $\checkmark$ | $\checkmark$ | $\checkmark$ | $\checkmark$ |
| CGenPU[27] | — | — | $\checkmark$ | — | — |

# D EXPERIMENTAL SETTINGS AND RESULTS

## D.1 EXPERIMENTAL SETTINGS

**Neural backbone architectures.** IMDb and 20News are handled by the MLP backbone summarised in Table D.2. Each document is first converted off-line into a 384-dimensional dense vector using the Sentence-Transformers all-MiniLM-L6-v2 model.[1] The vectors are stored in compressed NumPy format and loaded at training time, removing any text-processing overhead while guaranteeing identical representations for all PU learners. MNIST and F-MNIST share the LeNet architecture detailed in Table D.3. The network receives single-channel $28 \times 28$ images and produces one logit; when a method explicitly requires two outputs (e.g. HolisticPU variants) only the final linear layer is duplicated. CIFAR-10 uses the Custom CNN whose layer-wise definition is given in Table D.4. Designed for $32 \times 32$ three-channel inputs, the same "replace-last-layer" rule applies for methods that need 2-way logits. ADNI uses a CNN tailored for structural MRI, detailed in Table D.5; it receives single-channel $128 \times 128$ images and outputs one logit, and for methods requiring two outputs only the final linear layer is replaced. Spambase and Connect-4 are processed by the same MLP backbone shown in Table D.2. Spambase feeds the network with 57 raw numeric features, whereas Connect-4 first one-hot-encodes the board state into 126 binary features.

---

[1]https://huggingface.co/sentence-transformers/all-MiniLM-L6-v2

Table D.1: PN Train reports the standard supervised training split as $(N_p^{tr}, N_n^{tr})$. PU Train reports the PU training composition as $(N_l, N_u^+, N_u^-)$, where $N_l$ denotes the number of labeled positives, $N_u^+$ the number of positive instances present in the unlabeled pool, and $N_u^-$ the number of negatives in the unlabeled pool. Test uses $(N_p^{te}, N_n^{te})$. Validation shows the held-out fraction with its size $N_v$.

| Dataset | PN Train $(N_p^{tr}, N_n^{tr})$ | PU Train $(N_l, N_u^+, N_u^-)$ | Validation | Test $(N_p^{te}, N_n^{te})$ | Total Size |
|---|---|---|---|---|---|
| 20News | (6,326, 4,874) | (632, 6,326, 4,874) | 0.01 ($N_v = 114$) | (4,254, 3,278) | 18,846 |
| IMDb | (12,375, 12,375) | (1,237, 12,375, 12,375) | 0.01 ($N_v = 250$) | (12,500, 12,500) | 50,000 |
| MNIST | (29,197, 30,203) | (2,919, 29,197, 30,203) | 0.01 ($N_v = 600$) | (4,926, 5,074) | 70,000 |
| F-MNIST | (29,700, 29,700) | (2,970, 29,700, 29,700) | 0.01 ($N_v = 600$) | (5,000, 5,000) | 70,000 |
| CIFAR-10 | (19,800, 29,700) | (1,980, 19,800, 29,700) | 0.01 ($N_v = 500$) | (4,000, 6,000) | 60,000 |
| ADNI | (2,552, 2,516) | (255, 2,552, 2,516) | 0.01 ($N_v = 52$) | (622, 658) | 6,400 |
| Connect-4 | (35,222, 18,282) | (3,522, 35,222, 18,282) | 0.01 ($N_v = 541$) | (8,895, 4,617) | 67,557 |
| Spambase | (1,435, 2,208) | (143, 1,435, 2,208) | 0.01 ($N_v = 37$) | (363, 558) | 4,601 |

Table D.2: Layer-by-layer specification of the MLP backbone (IMDb / 20News / Spambase / Connect-4). The input feature dimension is $d \in \{384, 384, 57, 126\}$ depending on the dataset. Note: $p$ in Dropout$(p)$ denotes the dropout probability.

| # | Layer type | Output shape |
|---|---|---|
| 0 | Input | $d$ |
| 1 | Linear $(d \to 512)$ | 512 |
| 2 | ReLU | 512 |
| 3 | Dropout $(p = 0.3)$ | 512 |
| 4 | Linear $(512 \to 256)$ | 256 |
| 5 | ReLU | 256 |
| 6 | Dropout $(p = 0.3)$ | 256 |
| 7 | Linear $(256 \to 128)$ | 128 |
| 8 | ReLU | 128 |
| 9 | Dropout $(p = 0.2)$ | 128 |
| 10 | Linear $(128 \to 64)$ | 64 |
| 11 | ReLU | 64 |
| 12 | Linear $(64 \to 1)$ | 1 |

Table D.3: Layer-by-layer specification of the LeNet backbone (MNIST / F-MNIST). Note: Max-Pool2d uses $2 \times 2$ kernel with stride 2; Conv2d kernel sizes, strides, and padding are specified in the rightmost column.

| # | Layer type | Output shape | Kernel / stride / pad |
|---|---|---|---|
| 0 | Input | $(1, 28, 28)$ | – |
| 1 | Conv2d $(1 \to 10)$ | $(10, 24, 24)$ | $5 \times 5$, stride 1, pad 0 |
| 2 | MaxPool2d | $(10, 12, 12)$ | $2 \times 2$, stride 2 |
| 3 | ReLU | $(10, 12, 12)$ | – |
| 4 | Conv2d $(10 \to 20)$ | $(20, 8, 8)$ | $5 \times 5$, stride 1, pad 0 |
| 5 | MaxPool2d | $(20, 4, 4)$ | $2 \times 2$, stride 2 |
| 6 | ReLU | $(20, 4, 4)$ | – |
| 7 | Flatten | $(320, )$ | – |
| 8 | Linear $(320 \to 50)$ | $(50, )$ | – |
| 9 | ReLU | $(50, )$ | – |
| 10 | Linear $(50 \to 1)$ | $(1, )$ | – |

Table D.4: Layer-by-layer specification of the CIFAR-10 Custom CNN backbone. Note: $p$ in Dropout($p$) denotes the dropout probability; BatchNorm2d applies batch normalization; Conv2d parameters (kernel/stride/pad) are detailed in the rightmost column.

| # | Layer type | Output shape | Kernel / stride / pad |
|---|---|---|---|
| 0 | Input | $(3, 32, 32)$ | – |
| 1 | Conv2d $(3 \rightarrow 96)$ | $(96, 32, 32)$ | $3 \times 3$, stride 1, pad 1 |
| 2 | BatchNorm2d (96) | $(96, 32, 32)$ | – |
| 3 | ReLU | $(96, 32, 32)$ | – |
| 4 | Conv2d $(96 \rightarrow 96)$ | $(96, 32, 32)$ | $3 \times 3$, stride 1, pad 1 |
| 5 | BatchNorm2d (96) | $(96, 32, 32)$ | – |
| 6 | ReLU | $(96, 32, 32)$ | – |
| 7 | Conv2d $(96 \rightarrow 96)$ | $(96, 16, 16)$ | $3 \times 3$, stride 2, pad 1 |
| 8 | BatchNorm2d (96) | $(96, 16, 16)$ | – |
| 9 | ReLU | $(96, 16, 16)$ | – |
| 10 | Dropout $(p = 0.2)$ | $(96, 16, 16)$ | – |
| 11 | Conv2d $(96 \rightarrow 192)$ | $(192, 16, 16)$ | $3 \times 3$, stride 1, pad 1 |
| 12 | BatchNorm2d (192) | $(192, 16, 16)$ | – |
| 13 | ReLU | $(192, 16, 16)$ | – |
| 14 | Conv2d $(192 \rightarrow 192)$ | $(192, 16, 16)$ | $3 \times 3$, stride 1, pad 1 |
| 15 | BatchNorm2d (192) | $(192, 16, 16)$ | – |
| 16 | ReLU | $(192, 16, 16)$ | – |
| 17 | Conv2d $(192 \rightarrow 192)$ | $(192, 8, 8)$ | $3 \times 3$, stride 2, pad 1 |
| 18 | BatchNorm2d (192) | $(192, 8, 8)$ | – |
| 19 | ReLU | $(192, 8, 8)$ | – |
| 20 | Dropout $(p = 0.5)$ | $(192, 8, 8)$ | – |
| 21 | Flatten | $(12288, )$ | – |
| 22 | Linear $(12288 \rightarrow 1000)$ | $(1000, )$ | – |
| 23 | ReLU | $(1000, )$ | – |
| 24 | Dropout $(p = 0.5)$ | $(1000, )$ | – |
| 25 | Linear $(1000 \rightarrow 1000)$ | $(1000, )$ | – |
| 26 | ReLU | $(1000, )$ | – |
| 27 | Linear $(1000 \rightarrow 1)$ | $(1, )$ | – |

Table D.5: Layer-by-layer specification of the ADNI (Alzheimer MRI) CNN backbone. Note: $k$ in MaxPool2d($k$) denotes the kernel size ($k \times k$); $p$ in Dropout($p$) denotes the dropout probability; AdaptiveAvgPool2d$(1, 1)$ performs global average pooling.

| # | Layer type | Output shape | Kernel / stride / pad |
|---|---|---|---|
| 0 | Input | $(1, 128, 128)$ | – |
| 1 | Conv2d $(1 \rightarrow 32)$ | $(32, 128, 128)$ | $3 \times 3$, stride 1, pad 1 |
| 2 | BatchNorm2d $(32)$ | $(32, 128, 128)$ | – |
| 3 | ReLU | $(32, 128, 128)$ | – |
| 4 | Conv2d $(32 \rightarrow 32)$ | $(32, 128, 128)$ | $3 \times 3$, stride 1, pad 1 |
| 5 | BatchNorm2d $(32)$ | $(32, 128, 128)$ | – |
| 6 | ReLU | $(32, 128, 128)$ | – |
| 7 | MaxPool2d $(k = 2)$ | $(32, 64, 64)$ | $2 \times 2$, stride 2, pad 0 |
| 8 | Dropout $(p = 0.1)$ | $(32, 64, 64)$ | – |
| 9 | Conv2d $(32 \rightarrow 64)$ | $(64, 64, 64)$ | $3 \times 3$, stride 1, pad 1 |
| 10 | BatchNorm2d $(64)$ | $(64, 64, 64)$ | – |
| 11 | ReLU | $(64, 64, 64)$ | – |
| 12 | Conv2d $(64 \rightarrow 64)$ | $(64, 64, 64)$ | $3 \times 3$, stride 1, pad 1 |
| 13 | BatchNorm2d $(64)$ | $(64, 64, 64)$ | – |
| 14 | ReLU | $(64, 64, 64)$ | – |
| 15 | MaxPool2d $(k = 2)$ | $(64, 32, 32)$ | $2 \times 2$, stride 2, pad 0 |
| 16 | Dropout $(p = 0.1)$ | $(64, 32, 32)$ | – |
| 17 | Conv2d $(64 \rightarrow 128)$ | $(128, 32, 32)$ | $3 \times 3$, stride 1, pad 1 |
| 18 | BatchNorm2d $(128)$ | $(128, 32, 32)$ | – |
| 19 | ReLU | $(128, 32, 32)$ | – |
| 20 | Conv2d $(128 \rightarrow 128)$ | $(128, 32, 32)$ | $3 \times 3$, stride 1, pad 1 |
| 21 | BatchNorm2d $(128)$ | $(128, 32, 32)$ | – |
| 22 | ReLU | $(128, 32, 32)$ | – |
| 23 | MaxPool2d $(k = 2)$ | $(128, 16, 16)$ | $2 \times 2$, stride 2, pad 0 |
| 24 | Dropout $(p = 0.2)$ | $(128, 16, 16)$ | – |
| 25 | Conv2d $(128 \rightarrow 256)$ | $(256, 16, 16)$ | $3 \times 3$, stride 1, pad 1 |
| 26 | BatchNorm2d $(256)$ | $(256, 16, 16)$ | – |
| 27 | ReLU | $(256, 16, 16)$ | – |
| 28 | Conv2d $(256 \rightarrow 256)$ | $(256, 16, 16)$ | $3 \times 3$, stride 1, pad 1 |
| 29 | BatchNorm2d $(256)$ | $(256, 16, 16)$ | – |
| 30 | ReLU | $(256, 16, 16)$ | – |
| 31 | MaxPool2d $(k = 2)$ | $(256, 8, 8)$ | $2 \times 2$, stride 2, pad 0 |
| 32 | Dropout $(p = 0.3)$ | $(256, 8, 8)$ | – |
| 33 | AdaptiveAvgPool2d $(1, 1)$ | $(256, 1, 1)$ | – |
| 34 | Flatten | $(256, )$ | – |
| 35 | Linear $(256 \rightarrow 64)$ | $(64, )$ | – |
| 36 | ReLU | $(64, )$ | – |
| 37 | Dropout $(p = 0.3)$ | $(64, )$ | – |
| 38 | Linear $(64 \rightarrow 1)$ | $(1, )$ | – |

### D.2 PERFORMANCE UNDER THE CONVENTIONAL SETTING

#### D.2.1 FULL EXPERIMENTAL RESULTS

This subsection complements the main conventional results in Table 1. For each dataset, we use the dataset-to-backbone mapping in Appendix D.1 and the PU data specification in Table D.1. The training set size $N$ and class prior $\pi$ are fixed per dataset. PU training splits are generated with the same conventional configuration used for the main table. All learners are trained under the unified pipeline with identical optimizer and schedules defined in the YAML configurations, and only the classification head is adapted when a method requires two logits. We select the checkpoint with the best validation macro-F1 and report Accuracy, Precision, Recall, macro-F1, and AUC on the held-out test set.

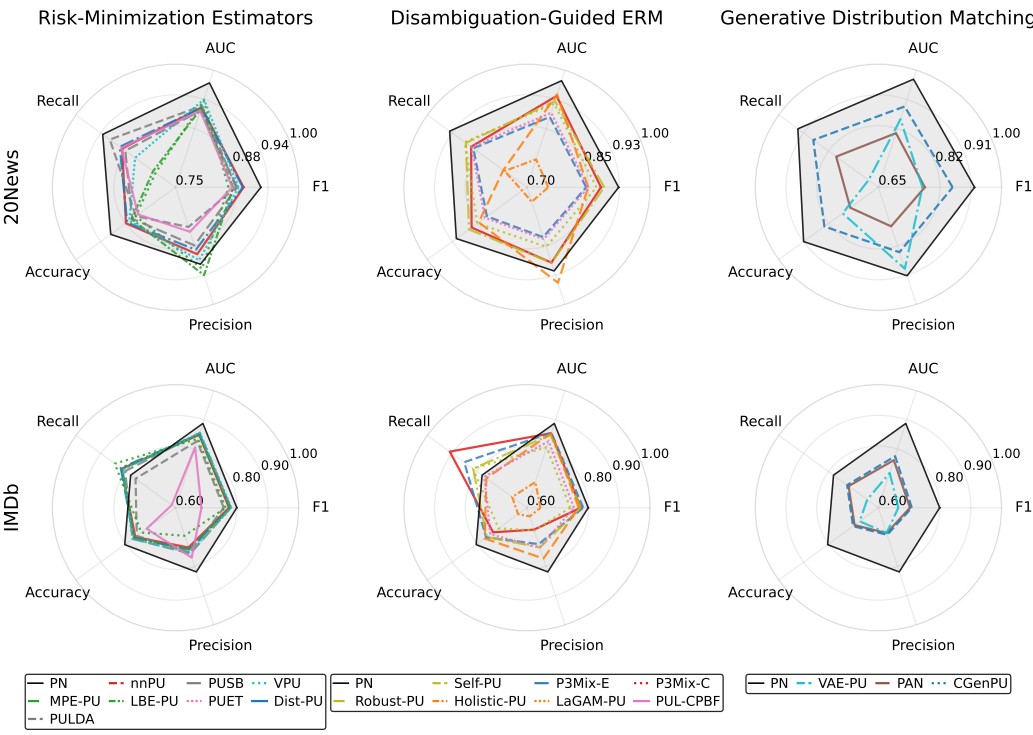

Figure 9: Performance of PU methods on text datasets across five evaluation metrics.

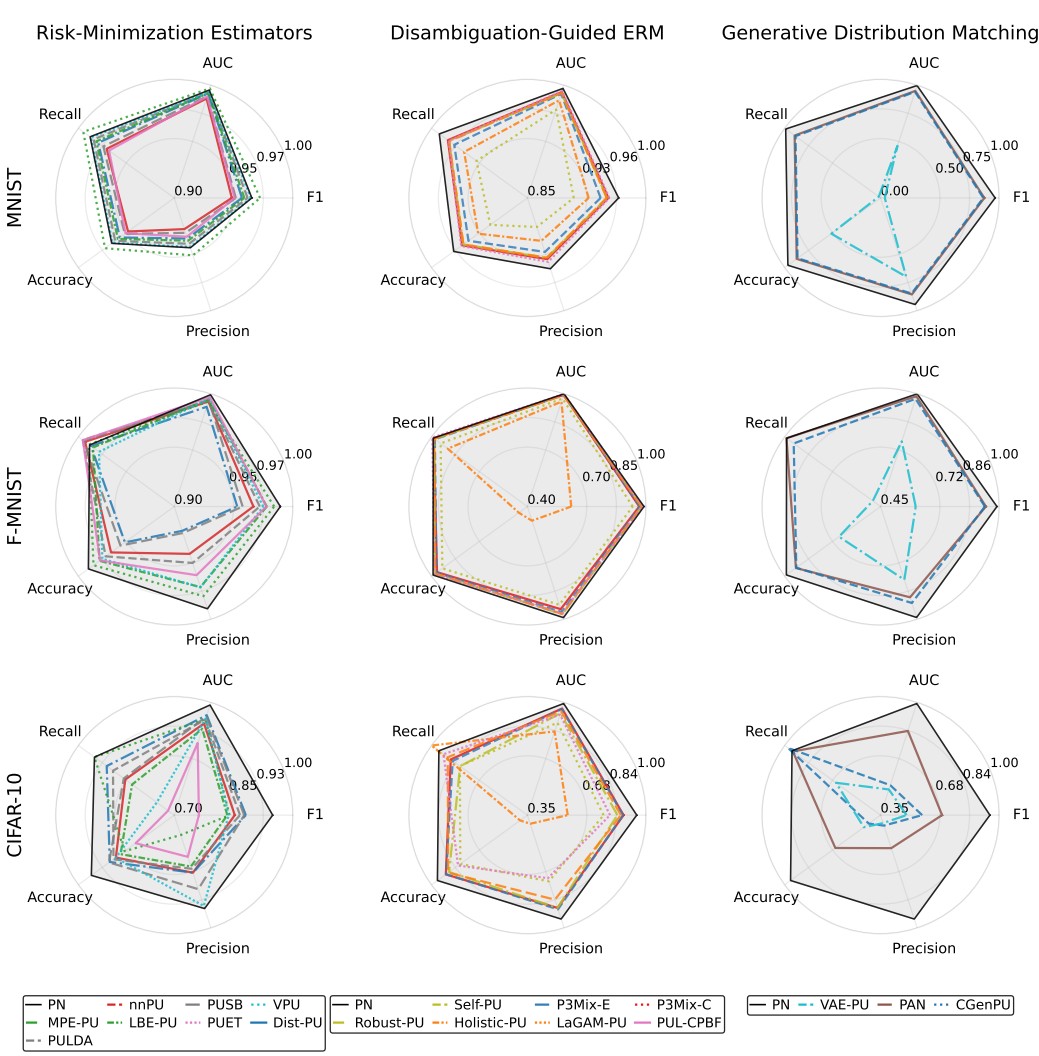

Figure 10: Performance of PU methods on vision datasets across five evaluation metrics.

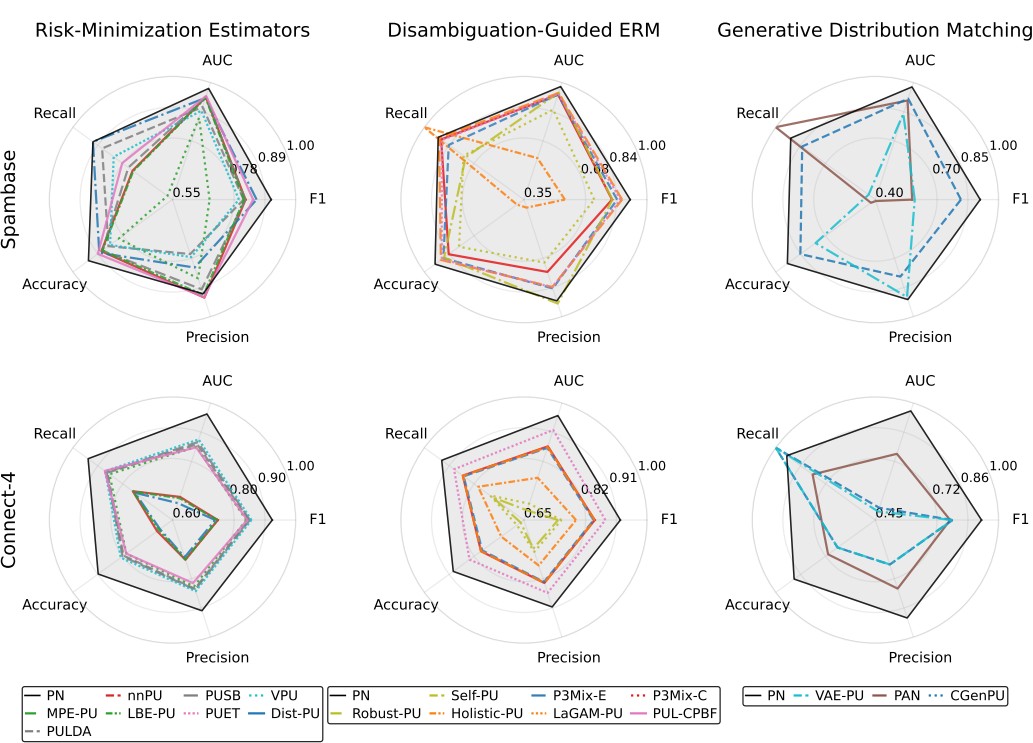

Figure 11: Performance of PU methods on tabular datasets across five evaluation metrics.

Table D.6: Holm–Bonferroni corrected $p$-values from two-sided paired $t$-tests on the test accuracies in Table 1 under the conventional setting (cc/SCAR/$c$=0.1). Each entry compares a PU method against nnPU on the same dataset. Values below $5e-02$ are shown in **bold** (significantly better than nnPU) or **underlined** (significantly worse).

| Method | IMDb | 20News | MNIST | F-MNIST | CIFAR-10 | ADNI | Connect-4 | Spambase |
|---|---|---|---|---|---|---|---|---|
| nnPU [15] | — | — | — | — | — | — | — | — |
| PUSB [14] | 8.2e-02 | 1.4e-01 | 9.5e-02 | 1.1e-01 | **4.3e-02** | 1.5e-01 | 1.6e-01 | 1.1e-01 |
| VPU [4] | **4.8e-02** | 5.8e-02 | **8.4e-05** | 8.7e-02 | 8.9e-02 | 8.5e-02 | **4.3e-02** | 9.2e-02 |
| MPE-PU [8] | 1.2e-01 | 1.6e-01 | 1.6e-01 | 2.0e-01 | 2.7e-01 | 2.3e-01 | 2.9e-01 | 2.1e-01 |
| LBE-PU [9] | 8.9e-02 | 1.5e-01 | **3.2e-03** | **2.1e-03** | 9.1e-02 | 6.9e-02 | 9.4e-02 | 8.3e-02 |
| PUET [30] | 9.5e-02 | 6.2e-02 | 9.8e-02 | 9.3e-02 | 6.4e-02 | **3.7e-02** | 8.1e-02 | 7.3e-02 |
| Dist-PU [36] | **3.9e-02** | 6.7e-02 | 8.7e-02 | 1.1e-01 | **8.9e-04** | **6.2e-03** | 7.6e-02 | **3.7e-02** |
| PULDA [13] | 1.8e-01 | 1.0e+00 | 8.1e-01 | 1.0e+00 | **1.3e-04** | 1.0e+00 | **5.9e-05** | 7.4e-02 |
| Self-PU [5] | **4.1e-02** | 1.8e-01 | 6.6e-02 | 2.0e-01 | 1.9e-01 | 1.9e-01 | 2.3e-01 | **3.8e-02** |
| P3Mix-E [19] | **3.2e-02** | 8.9e-02 | 9.3e-02 | 8.5e-02 | 6.7e-02 | 7.6e-02 | 9.4e-02 | 8.2e-02 |
| P3Mix-C [19] | **1.3e-02** | 5.5e-02 | 8.6e-02 | 9.7e-02 | 7.1e-02 | **2.0e-02** | 6.8e-02 | 7.3e-02 |
| Robust-PU [39] | 1.3e-01 | 8.9e-02 | 1.1e-01 | 1.5e-01 | 1.3e-01 | 1.4e-01 | 1.8e-01 | 1.6e-01 |
| Holistic-PU [32] | 9.0e-02 | **4.2e-02** | 7.5e-02 | **4.3e-02** | **3.4e-02** | 1.0e-01 | 5.9e-02 | 1.1e-01 |
| LaGAM-PU [21] | 6.7e-02 | 5.8e-02 | 8.5e-02 | 8.9e-02 | 8.3e-02 | 5.9e-02 | **7.4e-05** | **5.3e-05** |
| PUL-CPBF [20] | 7.5e-02 | 8.1e-02 | **3.2e-02** | **3.3e-02** | 9.5e-02 | 7.1e-02 | **4.1e-05** | **3.8e-02** |
| VAE-PU [25] | **3.6e-02** | 8.3e-01 | **4.8e-02** | 2.9e-01 | **2.7e-02** | **4.5e-02** | 3.9e-01 | 3.0e-01 |
| PAN [11] | 5.6e-02 | **2.8e-02** | 6.8e-02 | 9.1e-02 | 8.7e-02 | 1.1e-01 | 1.2e-01 | 8.9e-02 |
| CGenPU [27] | 9.1e-02 | 1.4e-01 | 9.5e-02 | 8.0e-02 | 1.0e-01 | 1.0e-01 | 7.8e-02 | 8.4e-02 |

### D.2.2 STATISTICAL SIGNIFICANCE OF PERFORMANCE DIFFERENCES

For each dataset and PU method, we collect the test accuracy with 10 random seeds (2, 25, 42, 52, 99, 103, 250, 666, 777, 2026) under the conventional cc/SCAR/$c$=0.1 configuration. We then perform two-sided paired $t$-tests comparing each method against nnPU on the same dataset.

**Paired $t$-test.** For each pair of methods (e.g., method $A$ vs. nnPU) on a given dataset, let $\{x_i\}_{i=1}^n$ and $\{y_i\}_{i=1}^n$ denote the test accuracies from $n = 10$ independent runs, where $x_i$ and $y_i$ are paired (same random seed). We compute the paired differences $d_i = x_i - y_i$, their sample mean $\bar{d} = \frac{1}{n}\sum_{i=1}^n d_i$, and sample standard deviation $s_d = \sqrt{\frac{1}{n-1}\sum_{i=1}^n (d_i - \bar{d})^2}$. The test statistic is

$$t = \frac{\bar{d}}{s_d/\sqrt{n}},$$

which follows a $t$-distribution with $n - 1$ degrees of freedom under the null hypothesis of no difference. The raw two-sided $p$-value is given by $p_{\text{raw}} = P(|T_{n-1}| \geq |t|)$, where $T_{n-1}$ denotes a $t$-distributed random variable with $n - 1$ degrees of freedom.

**Holm–Bonferroni correction.** To control for multiple comparisons across $m$ hypotheses (one per method vs. nnPU) on the same dataset, we apply the Holm–Bonferroni procedure. Let $p_{(1)} \leq p_{(2)} \leq \cdots \leq p_{(m)}$ denote the sorted raw $p$-values. For each hypothesis $j$ in sorted order, the corrected $p$-value is

$$p_{\text{corr}}^{(j)} = \min\left\{1, \max_{k=1,\ldots,j}\left[(m - k + 1) \cdot p_{(k)}\right]\right\}.$$

We reject hypothesis $j$ if $p_{\text{corr}}^{(j)} \leq \alpha$ (we use $\alpha = 0.05$ in all experiments).

Table D.6 shows that, under the conventional setting, several *Risk-Minimization Estimation* methods and *Disambiguation-Guided Supervised ERM* methods achieve statistically significant accuracy gains over nnPU on a non-trivial subset of datasets, whereas self-training and holistic variants more often suffer significant degradation. *Generative distribution matching* approaches rarely yield significant benefits and can even be significantly worse than nnPU, highlighting their instability in this regime. Overall, these significance tests confirm that the performance gaps observed in Table 1 are not driven by random seed noise.

### D.2.3 FAILURE CASES ANALYSIS

LBE-PU (Gong et al., 2021) exhibits competitive performance on vision datasets under conventional settings, and on simpler images it can match or surpass strong PN baselines. This effectiveness stems from its labeling-bias model $e(x)$ capturing meaningful variation in positive labeling mechanisms. On ADNI under the conventional setting, however, only a small fraction of positives are labeled ($|\mathcal{LP}| = 255$) while most remain unlabeled ($|\mathcal{U}_P| = 2552$ vs. $|\mathcal{N}| = 2516$). As shown in Fig. 12 (top panels), this extreme imbalance breaks LBE-PU in two ways. First, the classifier $h_\theta(x)$ concentrates labeled positives near 1.0 but assigns similarly high scores to many negatives, causing severe overlap between $\mathcal{U}_P$ and $\mathcal{N}$ in both training (panel a.1) and test distributions (panel a.2). Second, the estimated propensities $e(x)$ for labeled versus unlabeled positives exhibit substantial IQR overlap (panel a.3), indicating that the learned $e(x)$ deviates from the SCAR assumption and fails to model the true labeling mechanism. Together, these failures prevent LBE-PU from recovering a reliable decision boundary on ADNI.

While PAN (Hu et al., 2021) demonstrates effective performance on datasets such as MNIST and IMDb—where its adversarial training framework successfully promotes separation between positive instances and challenging unlabeled samples—it exhibits a distinct failure pattern on Spambase. As illustrated in Fig. 12 (bottom panels), this complementary failure mode stems from the Discriminator's role in defining reliability weights. Specifically, Panel b.3 reveals that the Discriminator systematically assigns higher scores to true negatives than to both labeled and unlabeled positives ($\sigma(D(x))_{\mathcal{N}} > \sigma(D(x))_{\mathcal{LP}}, \sigma(D(x))_{\mathcal{U}_P}$). Consequently, the reliability weights rl $= \sigma(D(x))$ become concentrated on unlabeled examples that resemble negatives. This skew causes the Recognizer to learn a decision function $h_\theta(x)$ that positions nearly all positive instances alongside a substantial portion of negatives above the default threshold of $0.5$ (panels b.1–b.2), resulting in exceptionally high recall but merely moderate precision under fixed-threshold evaluation.

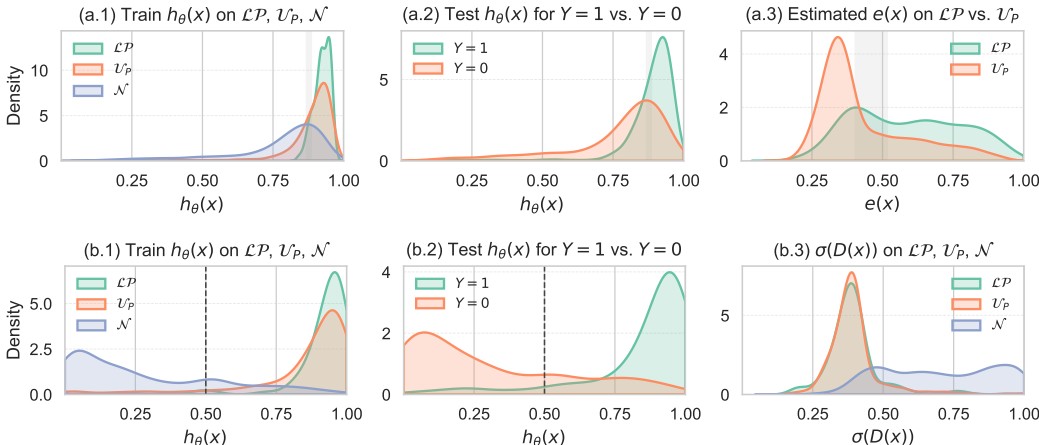

Figure 12: Failure analysis of LBE-PU on ADNI (top) and PAN on Spambase (bottom) under conventional settings. Top panels: (a.1, a.2) Training and test score distributions $h_\theta(x)$ for labeled positives $\mathcal{LP}$, unlabeled positives $\mathcal{U}_P$, and negatives $\mathcal{N}$; shaded bands indicate interquartile range (IQR) overlap between $\mathcal{U}_P$ and $\mathcal{N}$. (a.3) Estimated labeling propensity $e(x)$ for $\mathcal{LP}$ and $\mathcal{U}_P$ with IQR overlap highlighted. Bottom panels: (b.1, b.2) Training and test score distributions $h_\theta(x)$ with threshold at 0.5. (b.3) Discriminator scores $\sigma(D(x))$ on $\mathcal{LP}, \mathcal{U}_P$, and $\mathcal{N}$.

## D.3 EFFECTIVENESS OF PU METHODS W.R.T. LABEL RATIO

We study label efficiency by varying the label ratio $c$ while keeping the training size $N$ and class prior $\pi$ unchanged. Following the main text (Fig. 5), we sweep $c$ from 0.01 to 0.09 with a step of 0.02, and from 0.1 to 0.9 with a step of 0.1.

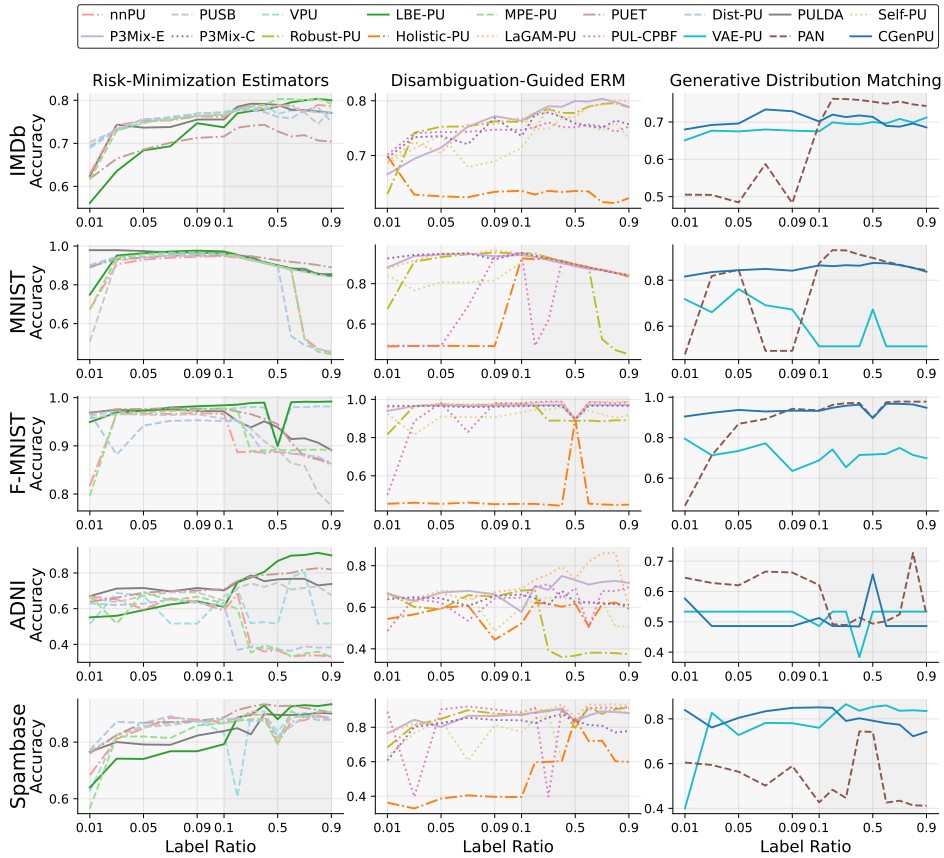

Figure 13: Effectiveness of PU methods with respect to different label ratios $c$.

## D.4 ROBUSTNESS TO SELECTION BIAS

We evaluate robustness under different labeling assumptions by varying the propensity of observing positive labels. The experiments follow the labeling strategies defined in Appendix C.3 (S1, S2, S3, S4). For each strategy, we keep the training size $N$ and class prior $\pi$ fixed. We report F1 on the test set at two label ratios, $c = 0.05$ and $c = 0.5$. Bars correspond to the low-label regime ($c = 0.05$) and lines correspond to the high-label regime ($c = 0.5$). All other training settings remain the same as in the conventional configuration.

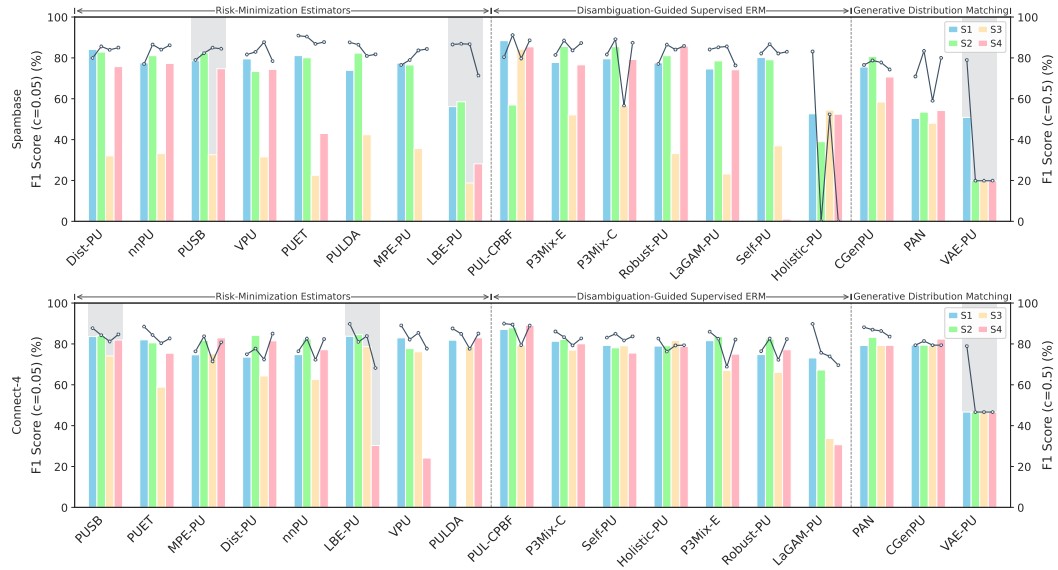

Figure 14: F1 scores of PU methods under different labeling assumptions on tabular datasets. Bars denote low-label regime ($c = 0.05$), lines denote high-label regime ($c = 0.5$).

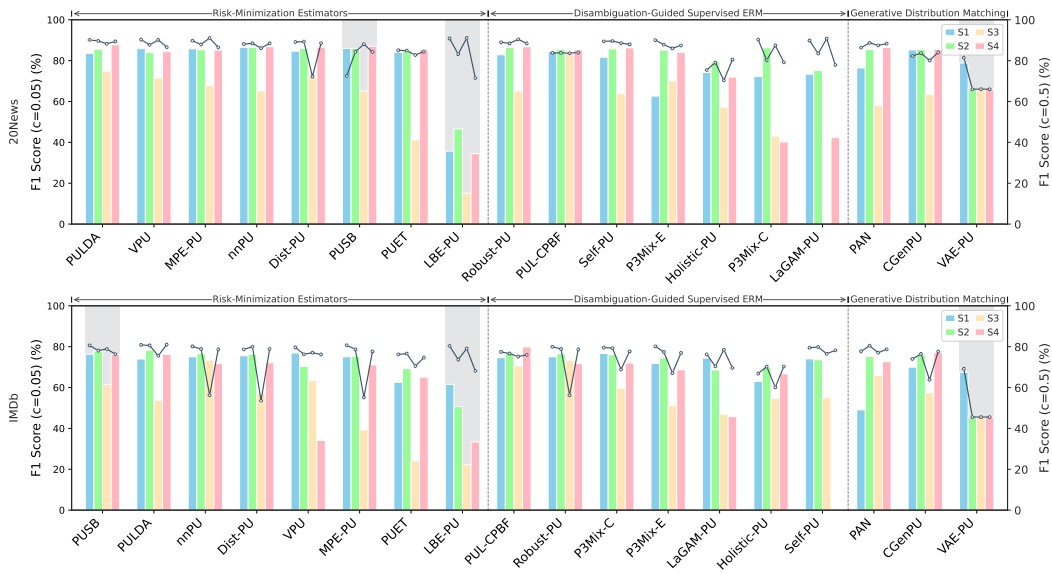

Figure 15: F1 scores of PU methods under different labeling assumptions on text datasets. Bars denote low-label regime ($c = 0.05$), lines denote high-label regime ($c = 0.5$).

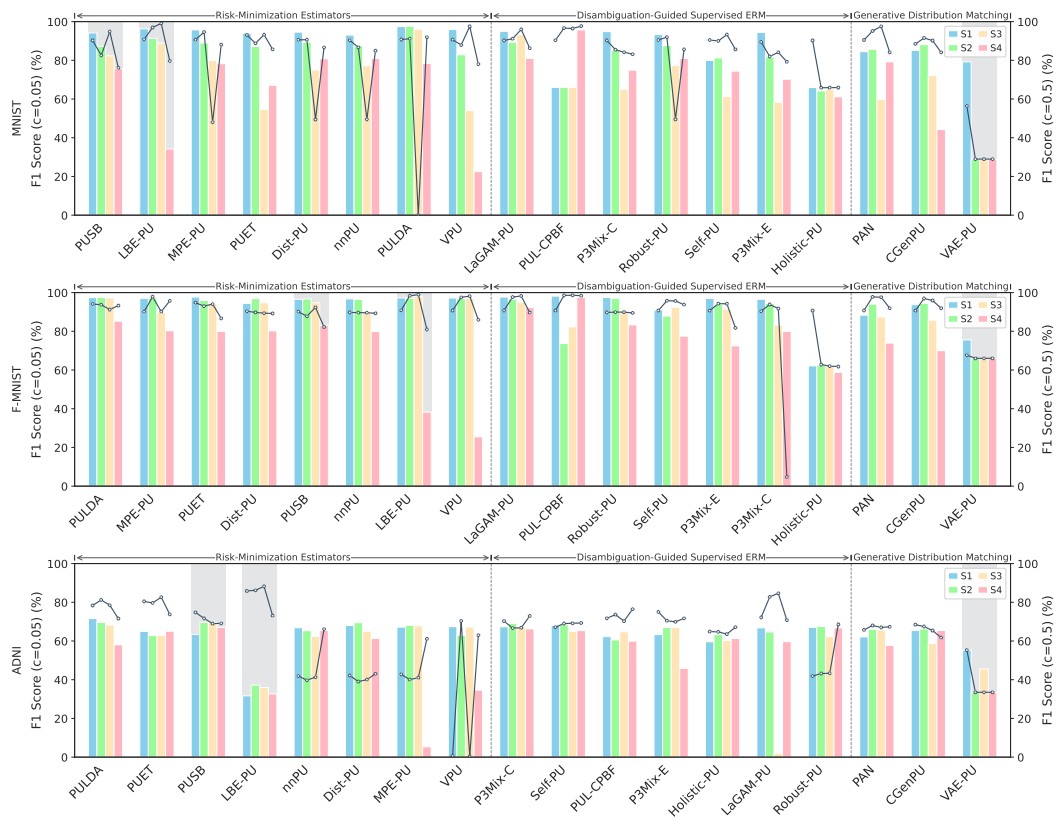

Figure 16: F1 scores of PU methods under different labeling assumptions on vision datasets. Bars denote low-label regime ($c = 0.05$), lines denote high-label regime ($c = 0.5$).

## E  COMPUTATION RESOURCES

All experiments were conducted using PyTorch 2.8.0 and Python 3.12 running on Ubuntu 22.04, with CUDA 12.8 for GPU acceleration. The hardware consisted of a single NVIDIA RTX 5090 GPU (32 GB memory), a 25 vCPU Intel Xeon Platinum 8470Q processor, and 90 GB of system RAM.

## F  PACKAGE, DOCUMENTATION, AND MAINTENANCE

This section outlines packaging, documentation, licensing, and long-term maintenance policies to support reliable use and extension of the benchmark.

**Data Acquisition.**  Datasets are obtained directly from authoritative sources through established APIs: vision (MNIST, F-MNIST, CIFAR-10) via torchvision; text (20News, IMDb) via scikit-learn and Hugging Face; tabular (Connect-4, Spambase) via OpenML; and medical imaging (AlzheimerMRI) via the ADNI database. Source-level acquisition lets users verify provenance and replicate exact conditions. Automatic downloading with caching reduces network overhead. Each loader performs integrity checks to verify data completeness and consistency before use. PU sampling is deterministic given the configuration and a random seed, which guarantees exact replication of splits and reported metrics when the same configuration is rerun.

**Documentation and usage.**  The package includes comprehensive documentation designed for PU learning workflows. It provides task-level examples and step-by-step instructions for construct-

ing PU datasets-supporting both SCAR and SAR mechanisms, as well as case-control and single-training-set scenarios. Users can specify class priors, label frequencies, model backbones, and PU learning objectives via YAML configuration files. Key components of the data generation pipeline, training loop, and evaluation suite are clearly structured and documented to facilitate adoption and extension.

**Code maintenance and versioning.** We are committed to maintaining the codebase, responding to user feedback, and encouraging community contributions through a structured review process. We adopt strict version control practices, maintain detailed changelogs, and provide tagged releases. Configuration files and experiment logs are archived alongside each release to ensure consistent behavior across versions and enable faithful replication when combined with the fixed-seed protocol described in the Reproducibility section.

**License.** Our code will be released under the MIT License, which permits free use, modification, distribution, and sublicensing, provided the original copyright notice and permission terms are retained.

## G    USE OF LARGE LANGUAGE MODELS (LLMS)

Large Language Models (LLMs) were employed for non-substantive, low-risk tasks in preparing this manuscript. Their use was strictly limited to grammar and typographical proofreading of human-drafted text and verifying the internal consistency of names and cross-references. To ensure scientific integrity and prevent any influence from hallucinated content, LLMs had zero involvement in generating, interpreting, or refining core scientific artifacts.

