# OpenReview forum: "PU-BENCH: A UNIFIED BENCHMARK FOR RIGOROUS AND REPRODUCIBLE PU LEARNING"
_ICLR.cc/2026/Conference — ICLR 2026 Poster_

### Official Review · Reviewer_eHTJ · 2025-10-17

**Soundness:** 2
**Presentation:** 2
**Contribution:** 3
**Rating:** 6
**Confidence:** 3

**Summary:**

This paper focuses on Positive-Unlabeled (PU) learning. It points out that although numerous PU learning methods have been proposed, the field still suffers from inconsistent data generation, disparate experimental settings, and divergent evaluation metrics, which have led to irreproducible results and unsubstantiated performance claims. Furthermore, the lack of a standardized and comprehensive benchmark has hindered rigorous evaluation. To address these issues, the paper introduces PU-BENCH, an open-source framework that provides a unified data generation pipeline and an integrated evaluation framework, aiming to catalyze reproducible, rigorous, and impactful research within the PU learning community.

**Strengths:**

1. The paper compiles and organizes a wide range of state-of-the-art PU learning methods, covering various algorithmic types and multiple public datasets. Moreover, for several advanced methods that were not publicly released, the authors conducted reimplementation efforts, which is commendable and worth encouraging.

2. Through extensive experiments, the paper clearly demonstrates the generality, robustness, and evaluation effectiveness of the proposed framework, highlighting the value of PU-BENCH in fostering reproducible and comparable research within the PU learning community.

**Weaknesses:**

1. In line 77, within the Data Sampling Scheme section, the description of the two sampling schemes is somewhat complex. It might be clearer and more intuitive if the authors could provide a simple illustrative example—such as showing how data are obtained under each sampling method—or include a figure or diagram to visualize the difference between the two schemes.

2. Although the paper presents a comprehensive benchmark for PU learning, including a unified data generation pipeline and an integrated framework, it lacks an analysis of the rationale behind these design choices. It would be helpful if the authors could explain why this particular configuration is reasonable and what criteria or considerations guided the design decisions.

3. The paper does not explain why these specific datasets were selected, which weakens the justification of the experimental setup. Providing reasoning for the dataset and method selection would make the evaluation more convincing.

**Questions:**

Please reply to my comments listed in "Weaknesses".

**Details Of Ethics Concerns:**

N/A.

---

> ### Author Response · Authors · 2025-11-18
>
> >Weakness #1: In line 77, within the Data Sampling Scheme section, the description of the two sampling schemes is somewhat complex. It might be clearer and more intuitive if the authors could provide a simple illustrative example—such as showing how data are obtained under each sampling method—or include a figure or diagram to visualize the difference between the two schemes.
>
> Thank you for this suggestion. We have added two figures in Appendix C.2 **(Lines 972-999 in the revised manuscript)** to better explain the two sampling schemes: **a schematic diagram (Fig. 7, Lines 972-984 in the revised manuscript)** contrasting the case-control vs. single-training-set constructions of $\mathcal{P}$, $\mathcal{N}$, $\mathcal{LP}$, and $\mathcal{U}$, and **a simple 1D Gaussian example (Fig. 8, Lines 985-999 in the revised manuscript)** illustrating how the resulting unlabeled density $f_u(x)$ differs.
>
> ---
>
> >Weakness #2: Although the paper presents a comprehensive benchmark for PU learning, including a unified data generation pipeline and an integrated framework, it lacks an analysis of the rationale behind these design choices. It would be helpful if the authors could explain why this particular configuration is reasonable and what criteria or considerations guided the design decisions.
>
> We appreciate your request and are pleased to provide further detail.
> Our unified data generation pipeline (**Lines 192-207 in the original submission**) is based on the key observation that there are **no standard "native" PU datasets**; all PU data are derived from fully-labeled classification datasets. Our pipeline explicitly models this derivation process by starting from fully-labeled datasets and sampling the Labeled Positives ($\mathcal{LP}$) based on three logical, core factors: (1) **data sampling scheme** (case-control vs. single-training-set), which determines where $\mathcal{LP}$ originates; (2) the **label frequency** $c$, which controls how many $\mathcal{LP}$ are sampled; and (3) the **labeling mechanism** (SCAR vs. SAR), which defines how $\mathcal{LP}$ is sampled.
>
> The architecture of our **integrated framework** (Figure 2, **Lines 162-182 in the original submission**) follows the traditional ML workflow: Data Preparation, Model Training, and Evaluation. Its modular design allows future researchers to easily integrate new datasets, incorporate new algorithms, or add new evaluation metrics, all while maintaining the benchmark's rigor and ensuring consistent, fair comparisons.
>
> ---
>
> > Weakness #3: The paper does not explain why these specific datasets were selected, which weakens the justification of the experimental setup. Providing reasoning for the dataset and method selection would make the evaluation more convincing.
>
> We appreciate the opportunity to clarify this point. The rational for the selection has already been explained in Section 3.1 ("Datasets and Methods", Lines 102-152). Our selection is principled:
> - The selected datasets (**Lines 104-124 in the original submission**): 1) cover diverse modalities (text, vision, tabular) and 2) are widely adopted in the literature, as summarized in Appendix Table C.1 (**Lines 919-942 in the original submission**).
> - The Methods (**Lines 132-138 in the original submission**): were selected based on: 1) Influence and Recency (prioritizing methods from "top-tier venues"), 2) Reproducibility (a strict requirement for "publicly available implementations"), 3) General Applicability (focusing on "domain-agnostic methods".

---

### Official Review · Reviewer_QcbA · 2025-10-31

**Soundness:** 3
**Presentation:** 3
**Contribution:** 3
**Rating:** 6
**Confidence:** 3

**Summary:**

The authors developed PU-Bench, an open-source framework that addresses the critical lack of standardized evaluation in PU learning research. This modular system includes a configurable data generator, unified training pipeline, and comprehensive evaluation suite. Extensive experiments are presented to compare 16 SOTA methods across datasets.

**Strengths:**

Through the results presented in this paper, the authors provides actionable insight for algorithm selection based on data modality, label availability and computational constraints. The key findings reveals valuable insight. No single method performs better universally, performance was highly dependent on the specific data type being used. The study also revealed clear trade-offs between predictive effectiveness and computational costs. All methods showed performance degradation when moving from random to biased labeling scenarios, though Risk-Minimization methods demonstrated greater robustness.
This work addresses fundamental challenges in PU learning research by establishing standardized evaluation protocols that enable fair method comparison and reproducible results.

**Weaknesses:**

1. The paper lacks statistical significance testing to determine whether performance differences between methods are meaningful or due to random variation. Results from single random seeds without multiple runs raise questions about reliability.
2. Also, fix data splits could affect the generalizability of findings.

**Questions:**

What is the variance in performance across different random initializations, and how does this affect method rankings?

---

> ### Author Response · Authors · 2025-11-18
>
> > Question: What is the variance in performance across different random initializations, and how does this affect method rankings?
>
> We appreciate you probing this critical experimental detail. Actually, the results reported in the original submission are averaged over ten independent runs with different seeds, not from a single seed. To prevent the ambiguity, we have updated table 1 (**Lines 217-232 in the revised manuscript**) with mean $\pm$ standard deviation.
>
> ---
>
> > Weakness #1 The paper lacks statistical significance testing to determine whether performance differences between methods are meaningful or due to random variation. Results from single random seeds without multiple runs raise questions about reliability.
>
> We share this concern about statistical reliability. For clarity, the results in our original submission were already averaged over ten independent runs as mentioned above. In response to this concern, we have  added a new subsection "D.2.2 Statistical significance of performance differences" in Appendix D (**Lines 1404-1457 in the revised manuscript**). This subsection reports $p$-values from two-sided paired $t$-tests, which confirm that our main conclusions are statistically significant and not due to random variation.
>
> ---
>
> > Weakness #2: Also, fix data splits could affect the generalizability of findings.
>
> We also agree with this point. As mentioned above, our experiments use different random seeds for each dataset, ensuring findings are not dependent on a single partition. Moreover, in the PU context, "data splitting" can also refer to how positive examples are assigned as labeled or unlabeled in the training dataset. We have evaluated this robustness across a wide range of label frequencies ($c$ = 0.01 to 0.9 in Section 5.1) and different labeling mechanisms (Section 5.2).

---

### Official Review · Reviewer_AKQF · 2025-11-01

**Soundness:** 3
**Presentation:** 3
**Contribution:** 3
**Rating:** 6
**Confidence:** 4

**Summary:**

This paper introduces PU-Bench, the first unified open-source benchmark for Positive-Unlabeled (PU) learning. It provides a standardized benchmark with a unified data generation pipeline, a framework of 16 state-of-the-art PU methods, and protocols for reproducible assessment. Through studies on 8 diverse datasets (2,560 evaluations), PU-Bench reveals trade-offs between effectiveness and efficiency, robustness and label frequency, and selection bias. This benchmark is meaningful for PU learning.

**Strengths:**

* The benchmark is well-designed with a unified data generation pipeline, a collection of state-of-the-art methods, and standardized evaluation protocols.
* The benchmark provides a more comprehensive setting and evaluation for PU learning, which is expected to promote the development of the field.
* The empirical study is extensive, covering multiple datasets and various conditions.

**Weaknesses:**

* While the benchmark itself is meaningful, the methods included are existing ones and lack certain theoretical analysis.
* Certain methods perform very well on some datasets but poorly on others. Additional visualizations and analyses of failure cases could provide valuable insights.
* The benchmark should be expandable, and experimental details can be specified via YAML. Some representative methods should also be compared [1-3].
[1] GradPU: Positive-Unlabeled Learning via Gradient Penalty and Positive Upweighting. AAAI2023
[2] Positive Distribution Pollution: Rethinking Positive Unlabeled Learning from a Unified Perspective. AAAI2023
[3] Positive-unlabeled learning with label distribution alignment. TPAMI2023

**Questions:**

See Weaknesses.

---

> ### Author Response · Authors · 2025-11-18
>
> >Weakness #1: While the benchmark itself is meaningful, the methods included are existing ones and lack certain theoretical analysis.
>
> Thank you for noting that "the benchmark itself is meaningful." We want to emphasize that the **motivation** for this work (**Lines 42-53 in the original submission**) stems from the observation that the current progress in PU learning has been "**systematically hindered by the lack of a standardized and comprehensive benchmark,**" leading to "**irreproducible findings and unsubstantiated performance claims.**" Therefore, our contribution is a rigorous empirical framework for existing methods, rather than a new, theoretically-analyzed algorithm. The contributions are:
> - The creation of PU-Bench: The first unified, open-source, and expandable framework for PU learning, which addresses a critical methodological gap.
> - A large-scale, systematic empirical analysis: Instead of a theoretical analysis of one method, our work provides an extensive empirical analysis of the entire field.
>
> We have added one sentence to our conclusion to more clearly frame our work's contribution as a foundational empirical study (**Lines 467-468 in the revised manuscript**).
>
> ---
>
> >Weakness #2: Certain methods perform very well on some datasets but poorly on others. Additional visualizations and analyses of failure cases could provide valuable insights.
>
> Thank you for this suggestion. Firstly, we would like to clarify that our study already identified the issue as a critical finding, as shown in Sec 4.1, "Performance is Highly Contingent on Data Modality," (Lines 305-312 in the original submission). Following your constructive suggestion, we have conducted a deeper analysis using score distribution histograms to elucidate the failure modes, as shown in **Lines 1458-1503 in the revised manuscript**.
> - **Visualization**: We log per-sample classifier scores together with PU and ground-truth labels, and visualize their distributions on both train and test splits (Figure 12).
> - **Analysis**: Two of the notable failure cases when changing datasets are 1) LBE-PU on ADNI, since its learned labeling-bias model $e(x)$ significantly deviates from the true SCAR assumption, and 2) PAN on Spambase, since the Discriminator biases reliability weights toward negative-like unlabeled instances (more details in **Lines 1481-1503 in the revised manuscript**).
>
> ---
>
> >Weakness #3.1: The benchmark should be expandable, and experimental details can be specified via YAML.
>
> We appreciate your recognition of these key aspects of PU-Bench’s design: the benchmark is extensible, and we indeed employed YAML to specify experimental configurations in the paper.
> - **Expandability**: PU-Bench is designed as a **fully modular, open-source framework** (Sec 3.2, Fig 2) specifically to be expandable. As detailed in Appendix F (Lines 1495-1515 in the original submission), the work will be released under an MIT license to encourage community contributions.
> - **YAML Configuration**: A core feature of our framework is its use of YAML for specifying experiment configurations. As shown in **Fig. 2 and detailed in Sec. 3.2**, this approach drives the entire experimental pipeline, i.e. controlling dataset selection, binarization, sampling, label ratios, model backbones, and hyperparameters. This ensures the rigorous reproducibility we identify as a key missing piece in the field.
>
> >Weakness #3.2: Some representative methods should also be compared [1-3].
>
> Thank you for raising this point. As we detailed in our original submission (**Lines 134-138 in the original submission**), a primary criterion for including a method in PU-Bench is the availability of a publicly released implementation. This is a crucial part of our commitment to ensuring code accuracy, consistency, and fair comparison. We have carefully investigated the three suggested publications based on this criterion:
> - For [3], we did find the corresponding public code. We are integrating this method [3] into PU-Bench and include its performance in our empirical study. The following shows some preliminary experimental results under the conventional setting (Accuracy \%):
> |Method|IMDb|20News|MNIST|F-MNIST|CIFAR-10|ADNI|Connect-4|Spambase|
> |-|-|-|-|-|-|-|-|-|
> |PULDA| $78.17 \pm 1.06$ | $87.82 \pm 2.21$ |$93.11 \pm 0.27$| $97.05 \pm 1.22$ | $88.95 \pm 0.67$ | $71.01 \pm 3.84$ | $86.61 \pm 1.27$ | $83.01 \pm 0.90$ |
> |PN| $79.89 \pm 0.83$ | $92.32 \pm 0.02$ |$96.54 \pm 0.92$ | $98.94 \pm 0.82$ | $94.88 \pm 0.57$ | $82.01 \pm 0.31$ | $92.38 \pm 0.78$ | $91.03 \pm 0.66$|
>
> - For [1] and [2], after a thorough search, we were unable to find publicly available, author-provided implementations for these two papers. To maintain the benchmark's high standard for code accuracy and fair comparison, we cannot include them in the experimental study at this time. We will, however, be sure to cite these three important papers in our related work section to provide a more complete context for the community.

---

### Author Response · Authors · 2025-12-01
**Overview of Revisions and Responses**

Dear Area Chairs and Reviewers,

Thank you for taking the time to review our submission and for your helpful comments. Your feedback helped us make the paper clearer and strengthen the experiments.

We are glad that the reviewers highlighted the following strengths of our work:

- **A well-designed unified framework** with a standard data generation pipeline and consistent evaluation protocols, which addresses fundamental challenges in PU learning (AKQF, QcbA).
- **An extensive collection of state-of-the-art methods** across various algorithmic types (AKQF, eHTJ).
- **Extensive empirical study covering multiple datasets and conditions** to demonstrate the framework's generality. The results provide **actionable insights** for algorithm selection, revealing critical findings regarding performance trade-offs and robustness (AKQF, QcbA, eHTJ).
- **PU-Bench shows significant value in fostering reproducible research within the community, and will actively promote further development in the field** (AKQF, QcbA, eHTJ).

In our rebuttal and revised manuscript, we **addressed the main concerns** as follows:

1. **Scope and contribution of the benchmark** (AKQF): We have updated the conclusion to state that our contribution is a rigorous empirical study to guide future theoretical and algorithmic advancements.

2. **Understanding method behavior and failure cases** (AKQF): **Score-distribution plots and detailed analyses** for representative failure cases have been added to the revised manuscript.

3. **Expandability and additional representative methods** (AKQF): We conducted **additional experiments** on the PU methods suggested by the reviewer.

4. **Statistical reliability and robustness** (QcbA): Table 1 has been updated with mean ± standard deviation values, and **paired t-tests** have been added to show that the main conclusions are statistically reliable and robust across different settings.

5. **Clarity of the data sampling schemes** (eHTJ): Two figures, a schematic diagram and a 1D Gaussian example, have been included in the revised manuscript to clearly illustrate the difference.

6. **Rationale behind the data generation pipeline, framework design, and dataset/method selection** (eHTJ): We emphasize that our design is guided by the alignment with existing PU practice, and the coverage of the most relevant experimental factors. The data generation pipeline is rooted in the core observation that **there are no standard "native" PU datasets**. Therefore, our pipeline explicitly models the data generation process by starting from fully-labeled datasets and sampling the labeled positives (shown in Figure 2). The whole framework follows a **standard ML workflow (data preprocessing, model training, evaluation)**, and our choice of **datasets and methods** is guided by modality diversity, common use in prior work, and the availability of public implementations.

We again thank the reviewers for their constructive comments and for helping us improve the work.

Best regards,

Authors of the ICLR 2026 submission 10731

---

### Meta-Review · Area_Chair_eppZ · 2026-01-06

**Summary:**

The reviewers generally agree that this paper presents a comprehensive and carefully engineered benchmark for PU learning, with standardized evaluation protocols and extensive empirical results across datasets and settings. At the same time, even where empirical observations are identified, they may not be sufficiently deep or impactful to move beyond performance-level benchmarking. Reviewers point out the absence of theoretical or principled analysis for the observed phenomenon.

**Reviewer Concerns:**

**Concerns addressed**

The rebuttal effectively addresses several coverage and clarity-related concerns raised by the reviewers. Specifically, the authors clarified the data generation and sampling with additional figures and examples, expanded experimental coverage, and additionally reported results over multiple runs and added statistical significance testing.

**Concerns still outstanding**

The core concern regarding the depth and significance of the insights derived from the benchmark remains outstanding. Specifically,
1.While the benchmark itself is meaningful, the methods included are existing ones and lack certain theoretical analysis.
2.... could provide valuable insights.

**Reviewer Scores:**

**Reviewer AKQF**
The rebuttal addresses several clarity and completeness issues, but does not resolve the reviewer’s concern regarding the lack of deeper theoretical or analytical insight. I expect the score would remain unchanged.

**Reviewer QcbA**
The rebuttal improves statistical rigorness and clarifies experimental reliability. But since the reviewer’s positive assessment was already marginal and the core contribution remains unchanged, I cannot expect a substantial increase in score.

**Reviewer eHTJ**
The rebuttal satisfactorily clarifies design choices and dataset/method selection rationale. Given that the reviewer’s evaluation focused primarily on design and completeness compared to contribution depth, the assessment would mostly remain similar.

---

### Decision · Program_Chairs · 2026-01-26

Accept (Poster)